# Non-Faradaic optoelectrodes for safe electrical neuromodulation

Jian Chen[1,7], Yanyan Liu [1,2,7], Feixiang Chen[1,7], Mengnan Guo[1], Jiajia Zhou [3], Pengfei Fu[2], Xin Zhang[2], Xueli Wang[4], He Wang [5], Wei Hua[2], Jinquan Chen [4], Jin Hu[2], Ying Mao [2] ✉, Dayong Jin [3,6] ✉ & Wenbo Bu [1,2] ✉

Nanoscale optoelectrodes hold the potential to stimulate optically individual neurons and intracellular organelles, a challenge that demands both a high-density of photoelectron storage and significant charge injection. Here, we report that zinc porphyrin, commonly used in dye-sensitized solar cells, can be self-assembled into nanorods and then coated by $TiO_2$. The J-aggregated zinc porphyrin array enables long-range exciton diffusion and allows for fast electron transfer into $TiO_2$. The formation of $TiO_2(e^-)$ attracts positive charges around the neuron membrane, contributing to the induction of action potentials. Far-field cranial irradiation of the motor cortex using a 670 nm laser or an 850 nm femtosecond laser can modulate local neuronal firing and trigger motor responses in the hind limb of mice. The pulsed photoelectrical stimulation of neurons in the subthalamic nucleus alleviates parkinsonian symptoms in mice, improving abnormal stepping and enhancing the activity of dopaminergic neurons. Our results suggest injectable nanoscopic optoelectrodes for optical neuromodulation with high efficiency and negligible side effects.

Electrical stimulation has been shown to be a useful method in the neuroscience and treatment of neural diseases[1–3]. Electrodes play a key role in mediating the transition from the electric current to ion flow at the electrode/tissue interfaces; this is essential for inducing membrane depolarization and initiating neurological function[4–7]. However, micropipette electrodes used in electrical stimulation are too large for cellular-level stimulations. Due to the mechanical instabilities and the cytosol dilution effect, neuroinflammation around the electrode often occurs after an electrode is implanted in the brain; this causes abnormal proliferation of glia, and the electrode becomes unsuitable for prolonged usage. Moreover, a higher stimulation voltage is typically needed to initiate the stimulation of nerves than early implantation due to gliosis after long implantation; this often triggers water decomposition and reactive oxygen generation in the tissue, causing a change in the microenvironment around the electrode that further damages the brain tissue.

Optogenetics has advanced the field by spatiotemporal precision; however, it requires the use of genetically encoded and optically active proteins to control neuronal activity, which limits its translational potential. Usually, the genetic modification process requires 2–4 weeks to express the target protein in neurons, thus increasing the cost of time and the probability of failure. Moreover, photosensitive proteins are activated by visible light at a power density of ~300–700 mW/cm², which may damage the nerve due to the thermal effect of the excitation light. Instead of electrode implantation, optical fiber implantation has typically

---

[1]Department of Materials Science, State Key Laboratory of Molecular Engineering of Polymers, Fudan University, Shanghai 200433, China. [2]Department of Neurosurgery, Huashan Hospital, Fudan University, Shanghai 200041, China. [3]Institute for Biomedical Materials and Devices (IBMD), Faculty of Science, University of Technology Sydney, Sydney, New South Wales 2007, Australia. [4]Sate Key Laboratory of Precision Spectroscopy, East China Normal University, Shanghai 200062, China. [5]Institute of Science and Technology for Brain Inspired Intelligence, Fudan University, Shanghai 200433, China. [6]Eastern Institute for Advanced Study, Eastern Institute of Technology, Ningbo, Zhejiang 315200, P.R. China. [7]These authors contributed equally: Jian Chen, Yanyan Liu, Feixiang Chen. ✉e-mail: maoying@fudan.edu.cn; dayong.Jin@uts.edu.au; wbbu@fudan.edu.cn

been needed due to the surgical complexity and mechanical concerns.

Nanotechnology-based on nanoscale materials, such as optoelectrodes, provides an exciting opportunity for neuromodulation, as nanodevices can be used in specific areas of the brain for far-field optically controlled electrical stimulation of cells; this provides precision for the control of neurons and even potentially subcellular organelles[4,8–12]. To implement this technology, the concept is to directly convert light into electricity. However, the challenge is also evident, as the commonly used charge-injection reactions are faradaic, where constant charge transfer occurs across the metal-solution interface and the electrode. Thus, it mostly involves irreversible redox processes encountered by water electrolysis or the reactive oxygen species formation, which are potentially harmful to the electrodes and tissues[13]. The non-Faradaic (capacitive) process, where a charge is progressively stored, is desirable, as it can reversibly charge and discharge at the electrode-electrolyte interfaces; however, a sufficiently high charge density is difficult to achieve, which typically exceeds $1 \, mC/cm^2$ [14,15].

Inspired by the dye-sensitized titania solar cells, the photoexcited electrons sensitized by dyes can be captured by $TiO_2$, forming $TiO_2(e^-)$ in an open circuit. $TiO_2(e^-)$ attracts cations (e.g., $Li^+$, $H^+$, $Na^+$) onto its surface, achieving ion current flow via a capacitive process[16,17]. Due to these features, $TiO_2$ is an ideal optoelectrode material dependent on the high-density photoelectron storage and non-Faradaic charge injection at the electrode-electrolyte interface. For the dye, its energetics must match that of $TiO_2$ for efficient $TiO_2(e^-)$ formation and that of reductants in tissue for significant dye regeneration. Porphyrin can be used for $TiO_2$ sensitization due to its large extinction coefficient, ultrafast electron injection, and slow charge recombination kinetics[18]. However, it often aggregates, which results in serious intramolecular exciton-exciton annihilation during exciton migration among the porphyrin units, thus reducing the efficiency of the electron supply for $TiO_2(e^-)$ formation.

In this work, we assemble zinc porphyrin (ZnTPyP) into J-aggregated nanorods to minimize exciton energy loss and enhance the photoelectric efficiency, as the self-assembled J-aggregated dyes in the nanostructures can lead to highly delocalized excited states, described as Frenkel excitons, for long-range energy transfer[19,20]. In our experiment, we construct a design of a ZnTPyP self-assembled nanorod coated by $TiO_2$ (ZST). The design of ZST nanoelectrodes enables either one-photon irradiation in the visible wavelength range or two-photon irradiation in the infrared spectrum range for deep tissue delivery of light. As shown in Fig. 1, ZST undergoes consecutive charge transfer reactions, including ultrafast exciton formation and migration to the heterojunction interface (~fs) and electron injection into $TiO_2$, to form long-lived (~ns) $TiO_2(e^-)$[21]. Upon 532 nm laser illumination, the charged photoelectrons can initiate ion redistribution near the cell membrane within several milliseconds for local membrane potential changes and action potential generation. Before injecting nanoelectrodes into the mouse motor cortex, spontaneous evoked neural firing was recorded upon 670 nm laser irradiation, confirming the ability of ZST to induce neuromodulation in vivo. Far-field cranial irradiation, either by a one-photon process at 670 nm or by a two-photon process using *fs* laser irradiation at 850 nm, induced the movement generation of mice at the hind limb, thus providing the advances of ZST in manipulating neural functions without the need for optical fiber implantation. Due to the capacitive nature and insufficient energetics of ZST, we confirmed that no toxic species were created.

## Results
### Controlled synthesis of ZST

We employed an acid–base neutralization reaction to initiate zinc meso-tetra (4-pyridyl) porphyrin (ZnTPyP) self-assembly into nanorods (ZS)[22,23] and yielded ZS with arbitrary lengths in the range of 200–1000 nm, which were suitable for the needed spatial resolution demand (Fig. S1a–c). We then redispersed the ZS nanorods in water with titanium diisopropoxidebis (acetylacetonate) (TDAA) for further coating of amorphous $TiO_2$, acquiring ZST with varied ratios of Zn/Ti (Fig. S1d–g). The X-ray diffraction (XRD) spectrum (Fig. S2a) and the corresponding simulated crystal structure of ZS (Fig. S2b) showed noncovalent π-π stacking between the ZnTPyP monomers, which was beneficial for exciton transport. XRD measurements of $TiO_2$ obtained

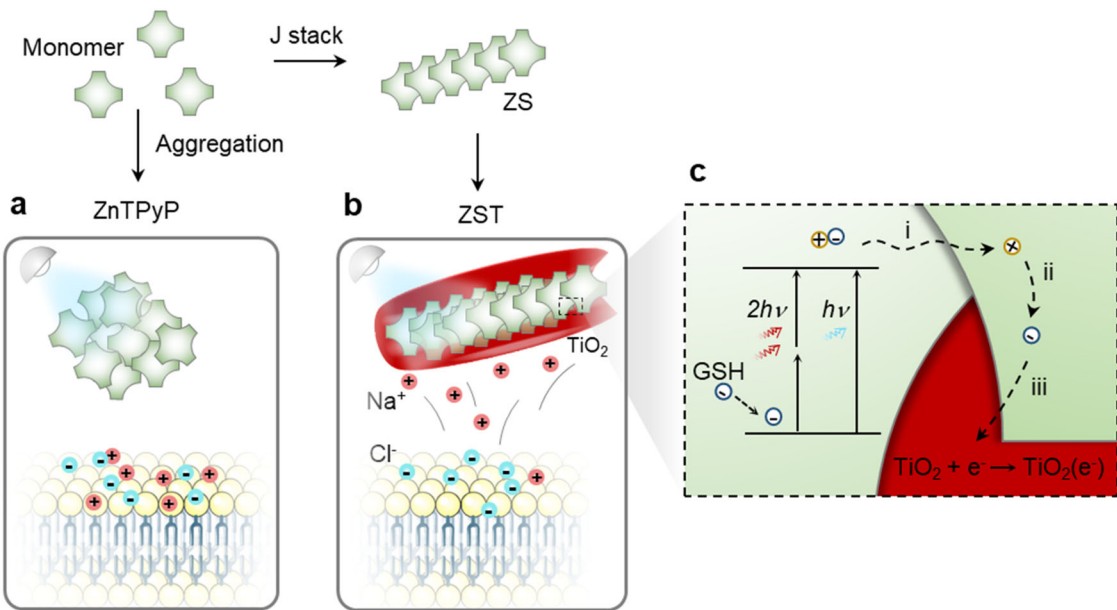

**Fig. 1 | The non-Faradaic capacitive mechanism of ZST for optoelectrical modulation of neuron. a** Random aggregation of ZnTPyP results in serious intramolecular exciton-exciton annihilation during exciton migration.
**b**, **c** J-aggregated ZnTPyP array enables the long-range exciton diffusion for ultrafast electron transfer into $TiO_2$, forming electron-rich $TiO_2(e^-)$. Meanwhile, the

co-generated holes are eliminated by the reductive scavengers in physiological environment like glutathione (GSH) for dye regeneration. i, exciton migration; ii, hole-electron pair separation; iii, electron transfer.

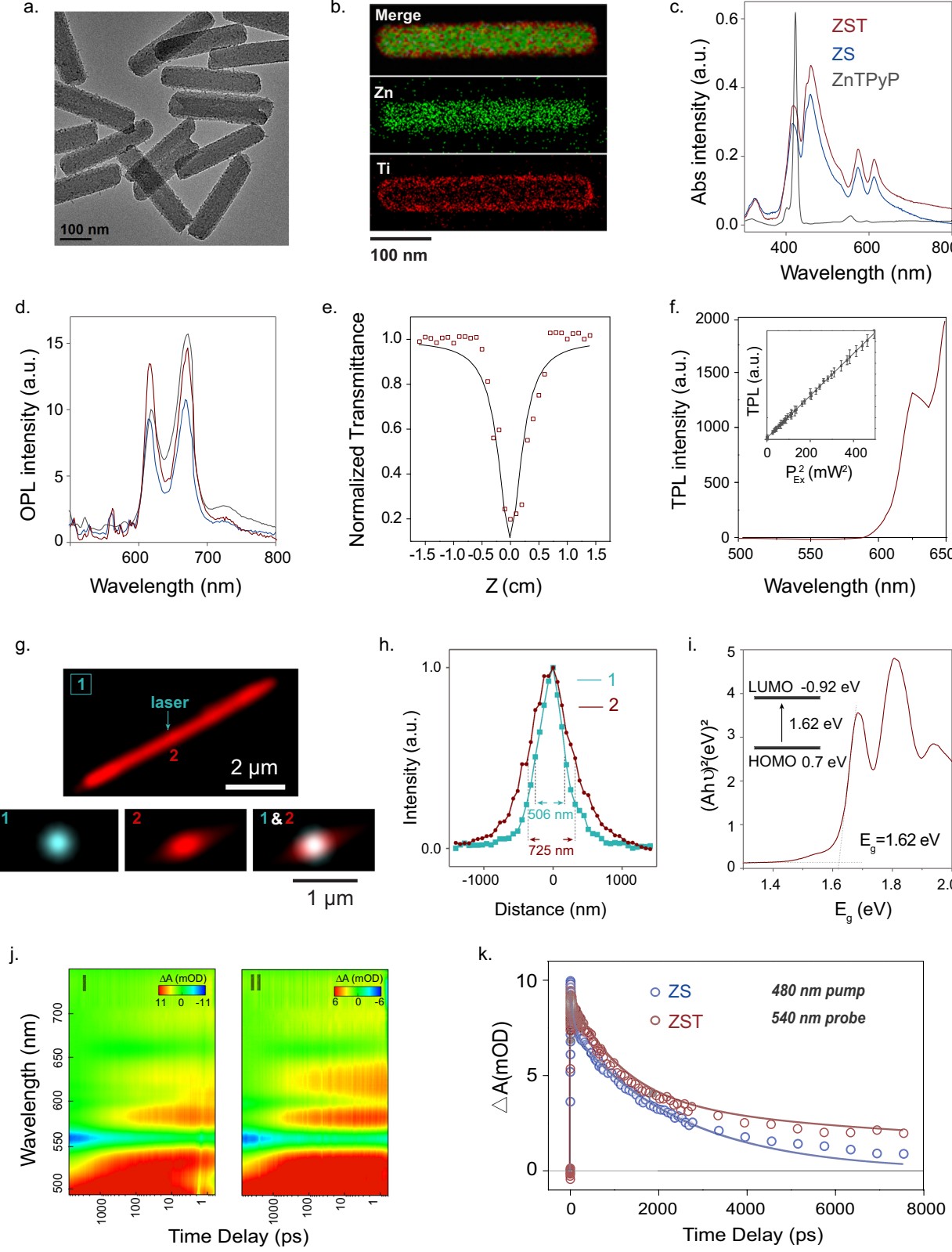

**Fig. 2 | Controlled synthesis and optoelectronics properties of ZST. a, b** TEM image (**a**) and EDS images (**b**) of ZST with Zn/Ti ratio of 1:1. **c** UV-vis absorption spectra of monomer ZnTPyP, ZS, and ZST. **d** PL spectra of ZnTPyP, ZS and ZST excited by 488 nm laser. **e** The open-aperture Z-scan signature of ZST at 850 nm fs laser. **f** TPL spectra of ZST excited by 850 nm fs laser. The insert image shows square dependence of luminescence on the excitation intensity. (The results are shown as mean ± SEM, $n$ = 3). **g** The reflected laser excitation beam (label 1) and the excitation on the local sample area (label 2). **h** The common Gaussian distributions of spot 1 and 2 in (**g**). **i** Tauc plots of ZS with the insert bandgap energy. **j** Transient absorption spectroscopy for (I) ZS and (II) ZST obtained after 480 nm fs laser excitation. **k** Transient absorption spectra detected at 480 nm pump and probe 540 nm for ZS and ZST. Data are representative of at least three independent experiments with similar results.

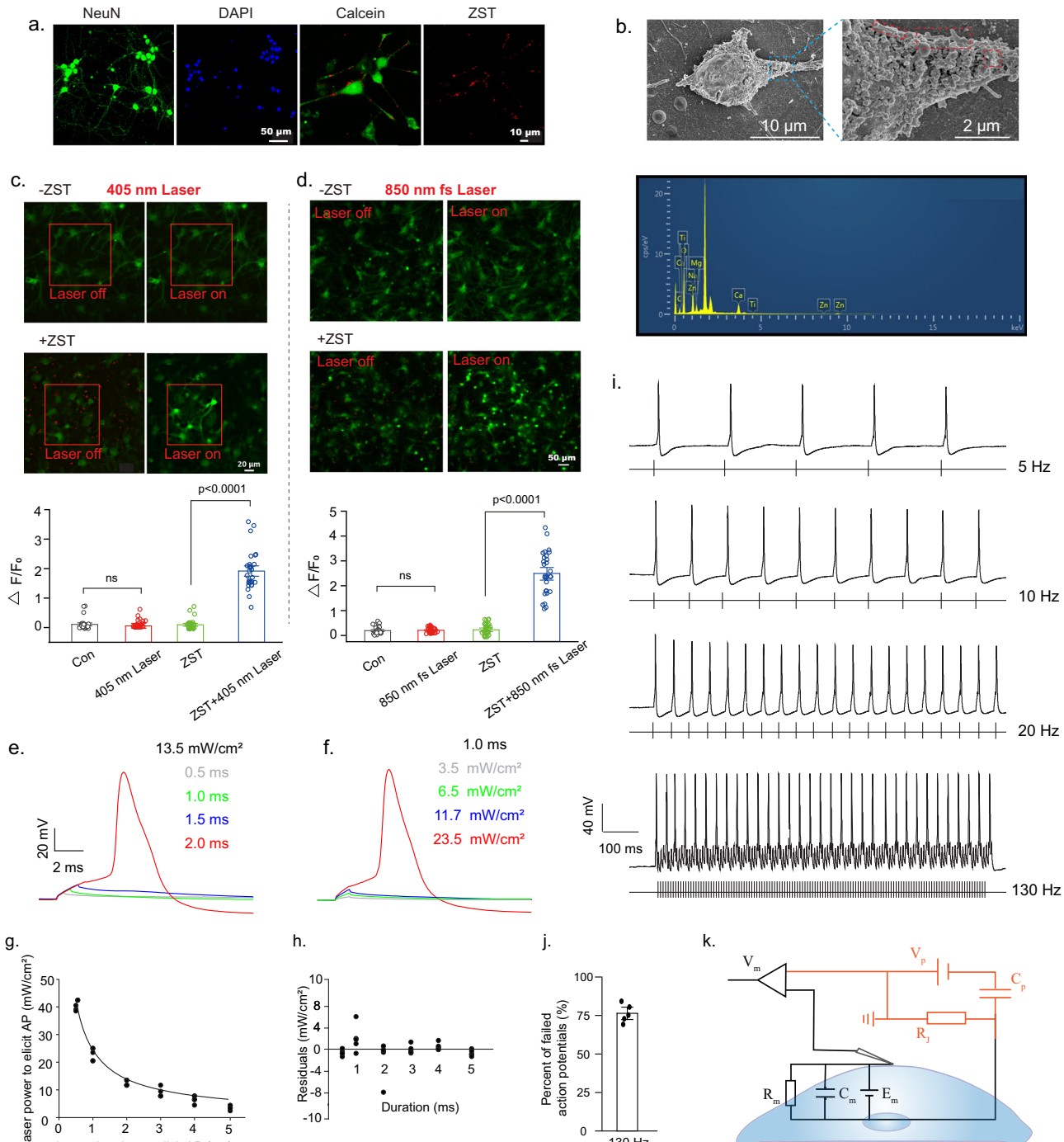

**Fig. 3 | Photocapacitive behavior and fast responses of ZST for membrane depolarization induction and action potential generation. a** NeuN and calcein immunostaining in rat cortical neurons. **b** SEM images and EDS spectra of ZST within the region outlined by the red dashed line on the outer membrane of neurons. **c** The $[Ca^{2+}]_i$ imaging in rat cortical neurons before and after 405 nm laser stimulation (the red box is the laser stimulation area), with the corresponding statistical fluorescence intensity of $[Ca^{2+}]_i$ in neurons pre-treated. (−Laser: no laser; +Laser: laser stimulation for 30 s; −ZST: no ZST; +ZST: ZST pre-treatment). **d** The $[Ca^{2+}]_i$ imaging in rat cortical neurons before and after 850 nm fs laser stimulation for 30 s. The results of **c** and **d** are shown as mean ± SEM by two-way ANOVA analysis followed by Bonferroni's post hoc test, $N = 30$ neurons. **e**, **f** The generation of AP on neurons pre-treated by ZST with different 532 nm laser power and pulse duration illumination. **g** An excitability curve displaying 532 nm laser power and

duration combinations that produce AP in neurons ($N = 5$ neurons; total of $N = 30$ replicates) to depict the amount of laser power required for AP generation. **h** Residual plot illustrating the goodness of fit of the hyperbolic fit curve to the data ($N = 6$ average laser powers for each laser duration used from 5 independent neurons and a total of 30 replicates). **i** AP firing under varied 532 nm laser stimulation frequencies. **j** the percentage of failed APs at 130 Hz that the neuron begins to fail to generate 1 AP per pulse of light from 5 independent neurons (The results are shown as mean ± SEM). **k** An equivalent stimulation circuit. $V_P$ and $C_P$ represent the photo-induced potential and capacitance of ZST; $R_J$ represents the ionic resistance between ZST and outer membrane; $E_m$, $C_m$, and $R_m$ represent transmembrane potential, capacitance, and resistance at resting state; $V_m$ is the measured membrane potential. Data are representative of at least three independent experiments with similar results.

via the above synthesis method confirmed its amorphous structure (Fig. S2c). ZST with a Zn/Ti ratio of 1:1 is shown in Fig. 2a, and the corresponding energy dispersive X-ray spectroscopy (EDS) elemental scanning images in Fig. 2b; additionally, the X-ray photoelectron spectroscopy (XPS) analysis showing two Ti $2p_{3/2}$ and $2p_{1/2}$ peaks at 464.8 eV and 458.9 eV in ZST in Fig. S2d further demonstrated the successful formation of $TiO_2$ on the ZS nanorod. Notably, the $TiO_2$ coating with the large presence of the surface O-H groups greatly improved the hydrophilicity of ZST, as evidenced by a smaller hydrodynamic particle size of 310.5 nm down from a size of 420.3 nm for ZS, along with a zeta potential decrease from −9.8 mV to −16.5 mV (Fig. S2e).

## One-photon and two-photon absorption properties of ZST

UV–vis absorption spectra showed that the intense Soret band (B-band) of ZnTPyP at 424 nm became split with a redshifted band arising at 462 nm for ZS and ZST, indicative of J-aggregation formation (Fig. 2c). The absorption peaks of ZST at approximately 600 nm were Q (0,0) and Q (1,0) from the Q-band. The appearance of the B-band and Q-band was attributed to transitions in the porphyrin ring, involving the $a_{1u}(\pi)$ $-e_g(\pi^*)$ and $a_{2u}(\pi)-e_g(\pi^*)$ electronic transitions[24,25]. Notably, the outer layer of $TiO_2$ had no significant effects on the ZS absorption property. One-photon luminescence (OPL) measurements showed two emission centers at 610 and 660 nm, corresponding to transitions from the material's first excited state and second excited state to the ground state[25] (Fig. 2d). Using the Z-scan technique, the nonlinear optical (NLO) properties of ZST were measured (Fig. 2e). As the distance between the ZST and the lens changed, the light transmittance showed an evident left-right symmetric valley shape, indicating the excellent NLO characteristics of the ZST using a two-photon absorption (TPA) cross section with values up to 2948 GM at 850 nm femtosecond pulsed laser excitation. The two-photon luminescence (TPL) spectral distribution (Fig. 2f) and the linear correlation between TPL intensity and the square of excitation intensity (the insert in Fig. 2f) confirmed the characteristics of a two-photon process. Due to the two-photon properties, ZST had great advantages of high-spatial resolution modulation of neurons as well as fluorescence tracing in vivo since infrared light could deeply penetrate through the tissues due to the low linear absorption and the scattering coefficient of biological specimens in the near infrared range.

## Exciton migration, separation and $TiO_2(e^-)$ formation within the capacitive ZST

To observe the exciton migration within the ZS nanorod, we used point laser illumination and wide-field imaging of the emission by a CMOS camera to record the spatial distribution of the energy migration along the microrod (Fig. 2g). A blank region (label 1) without a microrod was selected as a reference and showed a Gaussian distribution of the laser spot with a full width at half maximum (FWHM) of ~506 nm. In contrast, a profile with unambiguously oriented elongation along the microrod was observed with an FWHM of ~725 nm (label 2). The difference of ~219 nm in FWHM indicated that the excitons migrated along the microrod (Fig. 2h). From the density functional theory (DFT) calculations, the exciton binding energy ($E_b$) was 0.70 eV. Mott-Schottky analysis verified that ZS was a typical n-type semiconductor with an E-LUMO position at −0.92 V determined from the flat potential at −1.12 V (Fig. S3a), where the hot carriers flowed into the conduction band of $TiO_2$ for the photogenerated exciton separation and subsequent $TiO_2(e^-)$ formation. According to the Tauc plots (Fig. 2i), the bandgap energy of ZS was 1.62 eV, which became narrower in comparison to ZnTPyP, indicating enhanced visible light absorption[26]. This low energy level was insufficient to induce hydrolysis (3.4 eV) or generate hydroxyl radicals (1.9 eV) such that light irradiation of ZST did not cause any adverse effects on the neurons; thus, ZST was biologically safe during the photoelectric stimulation process. Based on the electrochemical impedance spectrum, the equivalent circuit model to

the circuit plot in the high-frequency region showed a capacitor parallel with an interface resistance, confirming the behavior of the electric double-layer capacitor of ZST (Fig. S3b).

## Fast photocurrent generation and enhanced photoelectron transportation from ZS to $TiO_2$

Using a patch-clamp setup and holding the voltage at zero to function as virtual ground in voltage-clamp mode, we observed the current generation by ZST (Fig. S4a, b) when the power density of 532 nm light illumination reached 13.5 mW/cm². Notably, under the same test conditions, the photocurrent peaks gradually increased as the Zn/Ti ratios decreased (see Fig. S4a) due to the efficient electron transfer at the ZS and $TiO_2$ heterojunction for a large current density output. As shown in Fig. S5, under 532 nm (23.5, 41.6 mW/cm²) and 670 nm (23.5, 41.6 mW/cm²) illumination, both the light source and ZST temperature changed less than 1 °C. To further elucidate this process, the transient absorption was measured with fs resolution. As shown in Fig. 2j, the excited state absorption of ZST (Zn/Ti ratio -1:1) in the range of 500 - 780 nm showed a significant time delay compared with that of ZS. From the kinetic traces at 540 nm, electron injection in ZST was observed to more quickly occur ($\tau_0 = 370 \pm 80$ fs) than that in ZS ($\tau_0 = 840 \pm 60$ fs), in accordance with the ultrafast electron injection kinetics of $TiO_2$ (Fig. 2k)[27]. After, a slower decay ($\tau_1 = 1199.1 \pm 107$ ps and $\tau_2 = 14238 \pm 5370$ ps) was observed in ZST than that in ZS ($\tau_1 = 85.92 \pm 8.11$ ps and $\tau_2 = 2490.3 \pm 63.1$ ps); this result strongly indicated enhanced photoelectron transport from ZS to $TiO_2$, leading to a decreased recombination probability.

Moreover, the co-generated holes could be eliminated by reductive GSH for dye regeneration (Fig. S4c). Electron paramagnetic resonance (EPR) showed the generation of superoxide anion ($O_2^{·-}$) during the photostimulation process of ZST (Fig. S4d), which was confirmed by superoxide anion kit detection, showing a large increase in $O_2^{·-}$ with the duration of laser illumination (Fig. S4e). The entire process was capacitive in photocapacitive mode, without water hydrolysis and hydroxyl radical (•OH) generation due to the insufficient photo electrochemical potentials of ZST, which were beneficial for its long-term neurostimulation (Fig. S4f).

## Safe in vitro neuromodulation with high-spatial resolution

Rat cortical neurons in primary culture displaying positive expression of neuronal nuclei (NeuN), a characteristic neuronal marker, were cultured for 15 days in vitro. The purity of the primary neuron cultures was assessed to be 75.8% ± 2.4% (Fig. S6a, b). The red fluorescence under 561 nm laser excitation indicates successful ZST adhesion onto the outer surface of neurons after 12 h of coculture (Fig. 3a). Scanning electron microscopy (SEM) and EDS further confirmed the close attachment of ZST on the neural membranes without internalization into cells (Fig. 3b). Cal-520 AM (AAT Bioquest), a fluorescent $Ca^{2+}$-detection probe, was used to stain the neurons for $Ca^{2+}$ imaging to visualize the neuronal activity. Upon 405 nm laser stimulation, sudden increases in $[Ca^{2+}]_i$ with spontaneous bursts of neuronal firing in the photostimulus region were clearly observed in the +ZST-pre-treated group (+Laser, +ZST). In contrast, light stimulation (+Laser, −ZST) or ZST (−Laser, +ZST) alone had no effect on the neural discharge and showed similar performance as that in the control group (−Laser, −ZST) (Fig. 3c). Notably, 850 nm fs laser stimulation also induced stimulated increases in $[Ca^{2+}]_i$ in +ZST-pre-treated neurons (Fig. 3d). Based on the above results, the output photocurrent of ZST upon either one-photon or two-photon irradiation successfully induced calcium influx for neuromodulation with high temporal and spatial resolutions. Moreover, no indication of phototoxicity was observed, as confirmed by the MTT assay results (Fig. S7a). Notably, no evident change in pH in the culture after light irradiation was observed, indicating the excellent biosafety of the ZST optoelectrodes (Fig. S7b).

## Minimal energy density in initiating action potential on a single-cell

The whole-cell current-clamp setup was used to record neural firing upon 532 nm laser illumination. With the laser power density at 13.5 mW/cm², neurons only produced subthreshold depolarizations at increasing pulse durations from 0.5 to 1.0 and 1.5 ms, and a pulse duration of 2.0 ms with an energy density of 27.0 µJ/cm² was found to be sufficient to initiate an AP (Fig. 3e). Similarly, when the pulse duration was set to 1.0 ms, an AP was produced until the laser power density reached 23.5 mW/cm², i.e., the energy density of 23.5 µJ/cm² (Fig. 3f). Thus, an excitability curve was simulated by a hyperbolic function to show the amount of laser power needed for AP generation (Fig. 3g, h). These results together showed that ZST could stimulate APs in a manner that was physiologically indistinguishable from those induced by classical external current injecting electrodes.

## Fast neuromodulation with high temporal resolution

We then chose to use a duration of 2.0 ms at a power of 13.5 mW/cm² to test different frequencies for neuron modulation. As shown in Fig. 3i, the photocurrent of ZST could enhance nerve excitation with millisecond temporal resolution, which was desirable for the precise control of neurons. The neurons were able to generate trains of APs when the optical stimulation frequency was modulated to up to 20 Hz. Notably, when the modulation frequency became too high (e.g., 130 Hz), the neurons could depolarize but not generate APs at each pulsed light, with approximately 76.4% ± 2.7% of the APs failing (Fig. 3j). This outcome occurred because ~80–90% of the cortical neurons cultured here were excitatory glutamatergic neurons with low firing frequencies in response to depolarizing stimulation. These neurons had membrane properties that caused them to be less sensitive to high-frequency stimuli. In contrast, ZS without TiO₂ coating could not initiate AP of neurons under the same light power density as that applied on ZST (Fig. S8).

An equivalent stimulation circuit is shown in Fig. 3k. Both the double-layer capacitor $C_p$ (at the ZST/electrolyte and electrolyte/neuronal membrane interface) and photogenerated potential $V_p$ dynamically depend on light illumination. In the dark, the ionic resistance between ZST and the outer membrane ($R_I$) is high, and the current is negligible. When the ZST is illuminated, $V_p$ is generated, and $C_p$ is immediately charged, resulting in a transient light-on current to mediate membrane depolarization for subsequent AP generation.

## In vivo neuromodulation by extracranial irradiation

ZST (28.7 µM in 2.0 µL) was injected into the motor cortex of adult male C57BL/6 mice, and the two-photon confocal image (Fig. 4a) shows the 3D distribution (red) of ZST within neurons (green). The bio-TEM image in Fig. 4b confirmed that ZST remained near the injection site in the brain for at least 30 days. The performance of ZST in the brain was evaluated by using in vivo multichannel electrophysiology by monitoring neuronal spiking activities. Far-field turning on the 670 nm laser resulted in significantly increased neural firing rates (+Laser, +ZST) (Fig. 4c, d). Notably, ZST remained effective after being injected into brain tissue for 30 days (Figure S10a, b). In contrast, ZST or laser alone had no significant effect. Moreover, with extracranial irradiation either by a one-photon process at 670 nm or by a two-photon process at 850 nm (Fig. 4e), we observed movement generation at the hind limb in mice with ZST pre-injected in the secondary motor cortex (M2) (+Laser, +ZST) (Fig. 4f–i, videos of hind limb movement under 670 nm laser stimulation in Supplementary Movie 1 and 850 nm fs laser stimulation in Supplementary Movie 2). In contrast, no detectable movement was observed in the control mice treated with ZST or laser alone. These results showed the potential of ZST in the in vivo modulation of the neural circuit functions, without the need for optical fiber implantation.

The mice were then sacrificed for immunostaining of c-fos expression[28]. A significant upregulation of c-fos was observed in mice with combined treatment using ZST and laser stimulation but not using ZST or laser stimulation alone; these results further confirmed the efficacy of ZST on the photoexcited neuronal activity at the molecular level (Fig. 4j, l)[29]. No distinct change in the number of astrocytes was observed in M2 30 days after stimuli treatments (Fig. 4k, m). We observed that the astrocytes did not undergo any changes in morphology across the different treatments, including surface area, number of processes, and total process length (Fig. S9a, b); additionally, the microglia did not exhibit any changes in morphology, including cell body size, number of processes, and total process length (Fig. S9c, d). These results showed the excellent biocompatibility of ZST and biosafety of this photoelectrical stimulation (Fig. 4k, m).

## Parkinson's disease therapy in mice

To study the therapeutic effects of ZST nanoelectrodes on the progression of PD in vivo, a reliable PD mouse model was further established by using the neurotoxin MPTP (1-methyl-4-fenyl-1,2,3,6-tetrahydropyridin)[30], with the flow chart of the therapeutic study illustrated in Fig. 5a. The C57BL/6 mice were intraperitoneally injected with MPTP every day for 7 days. An evident motor deficit was observed with the open-field test, characterized by decreased total movement distance ($p < 0.001$) and center distance ($p < 0.001$); this decrease was significantly correlated with the loss of striatal dopamine. For deep brain photoelectric stimulation, ZST (28.7 µM in 2.0 µL) was injected into the subthalamic nucleus (STN) with a fiber optic cable placed inside to couple with the 532 nm laser source. With a pulse width of 5 ms at 130 Hz, the pulse trains of 6 trials (each trial was composed of 10 s on and 20 s off durations) for a total of 3 min per day[31] were continually delivered for 10 days (Supplementary Dataset 1). After the treatment, the downregulation of total distance in the PD mice was markedly reversed. Notably, our treatment also ameliorated the depressive behavior of the PD mice, as evidenced by the increased total distance traveled in the center (Fig. 5b–d). This result was reasonable because depression is the most common psychiatric complication in PD that affects approximately half of PD patients and induces functional impairment. The gait variability of mice was also assessed by a computer-assisted gait analysis system, Catwalk. The results showed that MPTP-challenged mice had significant gait deficits in dynamic paw function and posture stability. Abnormal gait persisted over the three-week study period in the PD group. In contrast, ZST combined with laser treatment significantly improved the gait functions with an increase in the swing speed and a decrease in the variation of the stand time and step cycle (Fig. 5e, f).

Notably, this high-frequency photoelectric treatment by ZST switched off the pathologically disrupted activity in the STN; thus, it significantly alleviated motor symptoms (Fig. 6a). Photomicrographs of representative mouse brains showed the distribution of ZST in the STN 19 days after injection (Fig. 6b). Studies have shown that following the impairment of substantia nigra dopamine neurons, the STN also exhibits hyperactivity and abnormal rhythmic burst firing[32]. Therefore, the expression of abnormal firing may cause the alterations in PD's neural network. Here, the impact of dopaminergic neuron loss on STN synaptic transmission was measured by whole-cell patch-clamp recording. No significant impact on the spontaneous excitatory postsynaptic current (sEPSC) was observed (see Fig. 6c, d). In contrast, the frequency but not the amplitude of the spontaneous inhibitory postsynaptic current (sIPSC) amplitude was significantly increased by the dopamine depletion and even became higher after photoelectric treatment by ZST (Fig. 6e, f). This occurred because the increased inhibitory synaptic transmission via an "indirect" pathway could result in a decrease in STN neuron autonomous firing activity to prevent hyperactivity and abnormal rhythmic burst firing in STN[33]. These adaptations could reflect homeostatic compensatory processes

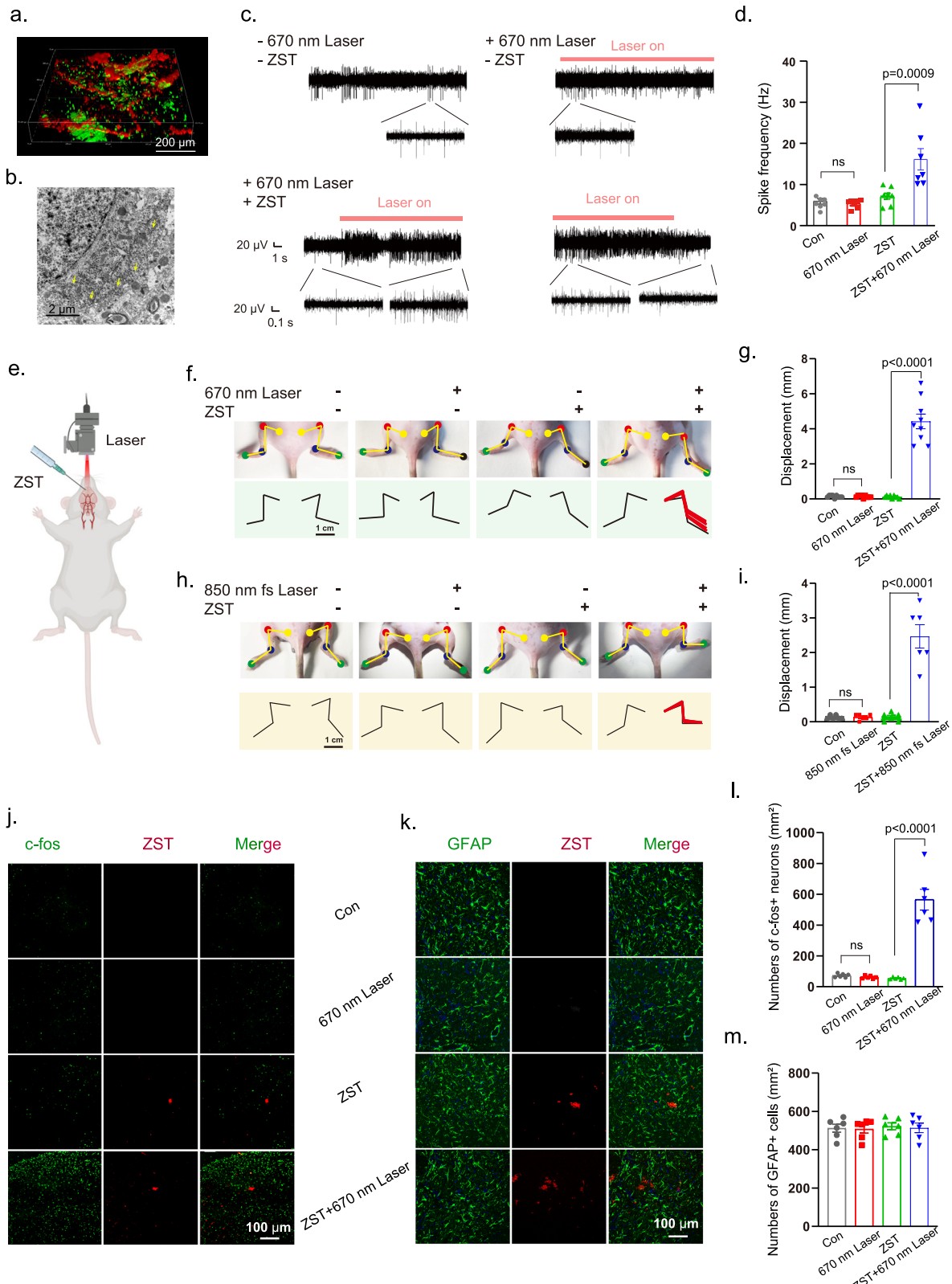

triggered by hyperactivity of striatal D2 dopamine receptor-expressing medium spiny neurons when DpA neurons are lost[33]. Studies show that a higher stimulation frequency correlates to a higher percentage of neurons with an inhibitory response for the reduction of STN activity[34]. Here, after photoelectric treatment by ZST, the frequency of sIPSC was markedly increased and could be more adaptive by contributing to normal firing patterns in the STN.

The immunohistochemistry results showed an evident decrease in TH expression in the substantia nigra pars compacta (SNc) of MPTP-induced PD mice, and this damage was significantly rescued after cotreatment with ZST+Laser with the upregulation of c-fos expression (Fig. 6g). The MPTP treatment produced a mean TH neuron loss of 62.8% compared to the PBS-treated control group; however, the ZST+Laser treatment resulted in an approximately 109.4% increase in TH

**Fig. 4 | In vivo neuromodulation by extracranial one-photon or two-photon laser stimulation. a** The 3D two-photon confocal fluorescence imaging of ZST (red fluorescence) and neurons (green fluorescence) in the mice genetically-encoded by GCaMP6 probes, with ZST pre-injected in M2. **b** Bio-TEM image of ZST distribution in tissues 30 days later after ZST was injected into brain (the yellow arrows point to ZST). **c, d** In vivo multichannel electrophysiology used to monitor the neuronal firing frequency of mice under different treatments. (−Laser: no laser; +Laser: laser irradiation; −ZST: no ZST; +ZST: ZST pre-injected in M2). The results are shown as mean ± SD by two-way ANOVA by Bonferroni's post hoc test, $N = 7$ neurons from 3 mice in each group. **e** The scheme of mice movement test (Created with BioRender.com.). **f, g** The movement at hind climb in mice with or without ZST pre-injected in M2, combined with or without extracranial 670 nm laser stimulation.

The results are shown as mean ± SD by two-way ANOVA by Bonferroni's post hoc test, $N = 3$ mice. **h, i** The movement at hind climb in mice with or without ZST pre-injected in M2, combined with or without extracranial 850 nm fs laser stimulation. The results are shown as mean ± SD by two-way ANOVA analysis followed by Bonferroni's post hoc test, $N = 3$ mice. **j** The immunostaining of c-fos expressed in M2 that was collected 2 hours later after different treatments. **k** The immunostaining of GFAP expressed in mice M2 that was collected 30 days after different treatments. The corresponding statistical numbers of neurons with (**l**) c-fos expression (c-fos+) in (**j**), and (**m**) GFAP expression (GFAP+) in (**k**). The results are shown as mean ± SD by two-way ANOVA analysis followed by Bonferroni's post hoc test, $N = 6$ mice in each group.

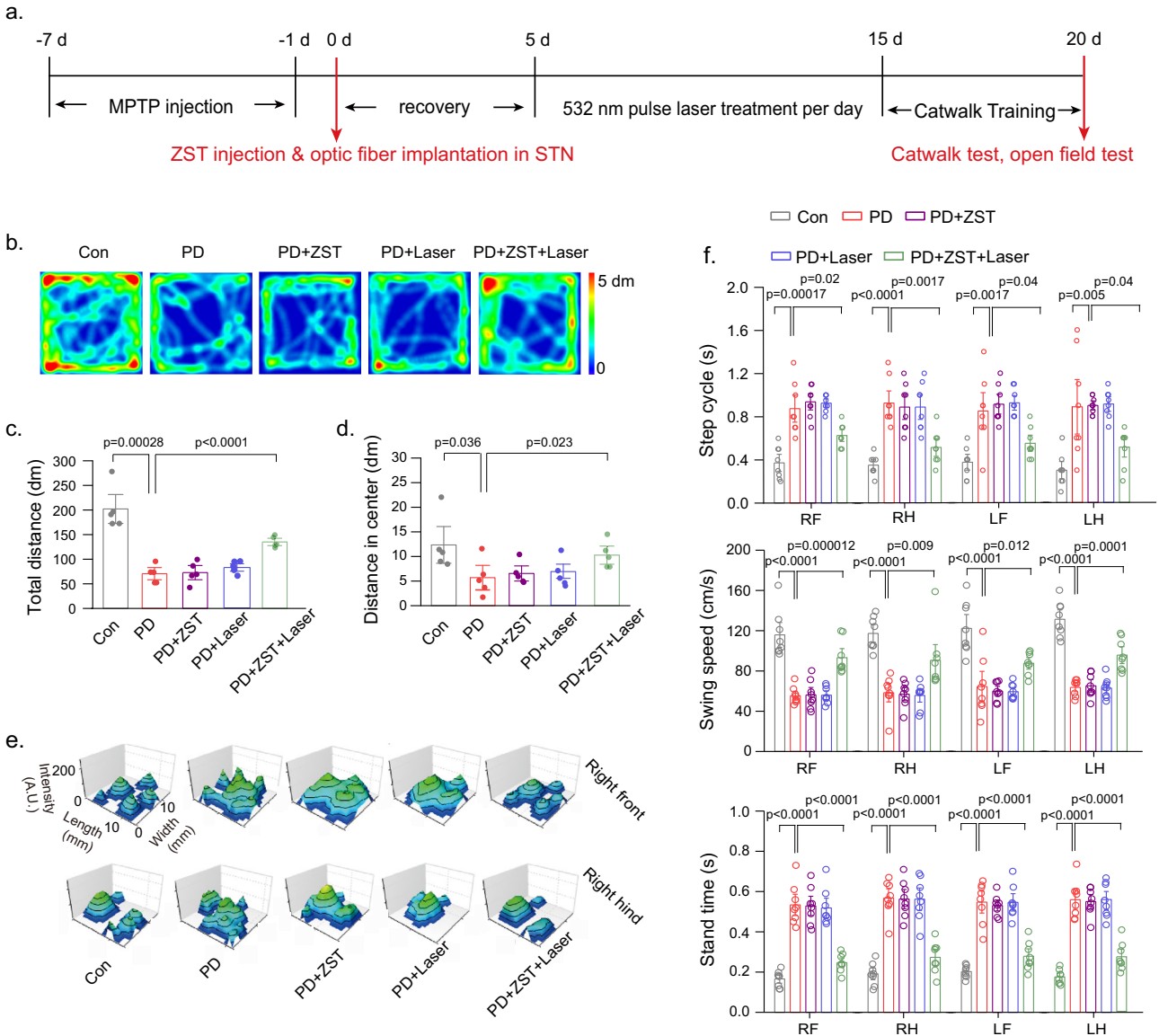

**Fig. 5 | Parkinson's Disease therapy in mice. a** Flow chart of the treatment study of MPTP-induced PD mice. **b** Representative trajectories and **c, d** locomotor activities in open-field test for mice in Con, PD, PD + ZST, PD+Laser, and PD + ZST+Laser. Laser wavelength is 532 nm. The results are shown as mean ± SEM by one-way ANOVA analysis followed by Bonferroni's post hoc test for Con and PD, $N = 5$ mice in each group. The results are shown as mean ± SEM by two-way ANOVA by Bonferroni's post hoc test for PD, PD + ZST, PD+Laser, and PD + ZST+Laser, $N = 5$ mice in

each group. **e, f** Representative illumination of footprints, and gait variability for mice in different treatment groups (RF = right forepaw, LF = left forepaw, RH = right hindpaw, and LH = left hindpaw). The results are shown as mean ± SEM by one-way ANOVA analysis followed by Bonferroni's post hoc test for Con and PD, $N = 8$ mice in each group. The results are shown as mean ± SEM by two-way ANOVA analysis followed by Bonferroni's post hoc test' for PD, PD + ZST, PD+Laser, and PD + ZST +Laser, $N = 8$ mice in each group.

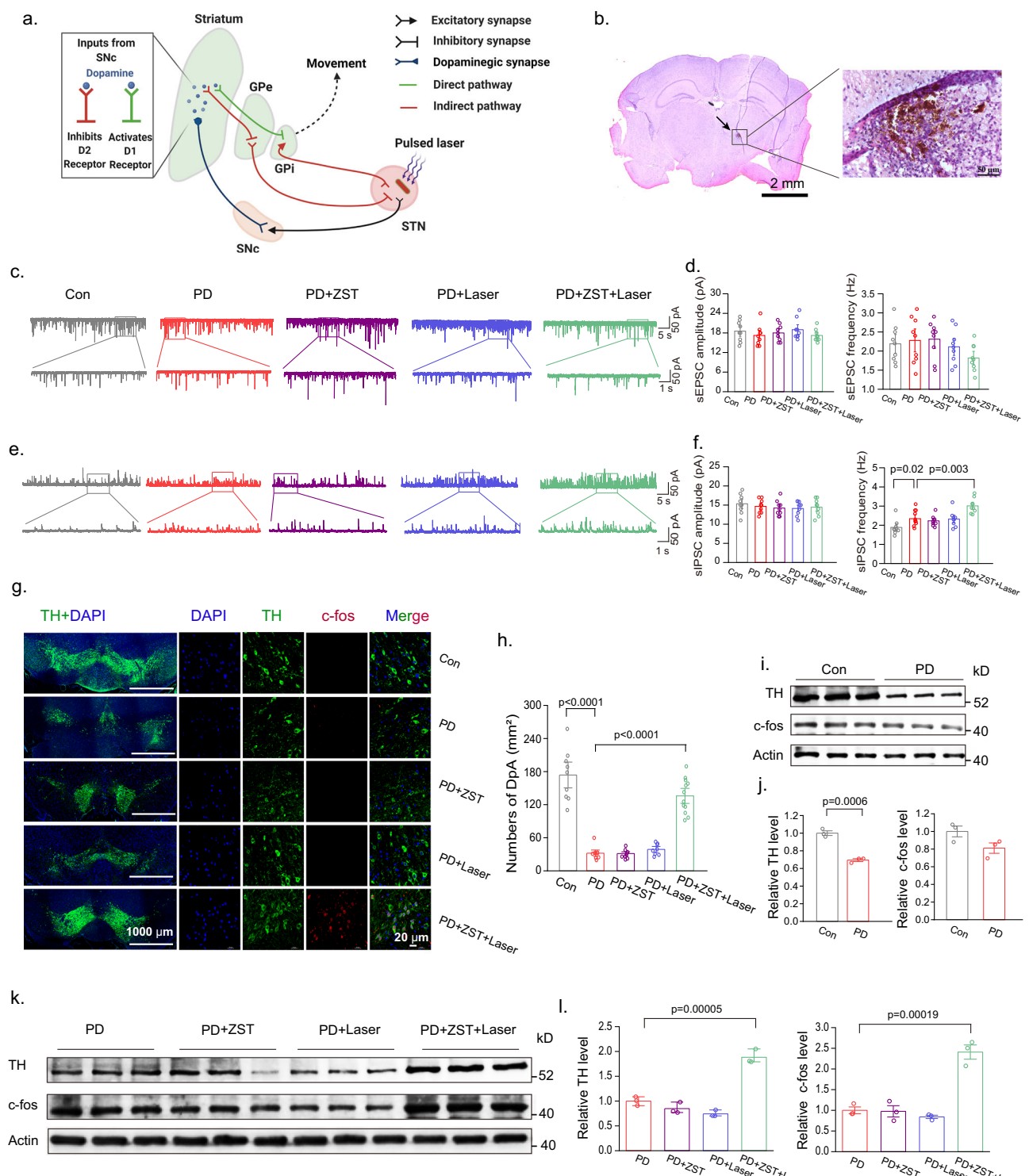

neurons in PD mice via stereological analysis (Fig. 6h). The Western blot analysis results in Fig. 6i, j further provided evidence of the positive therapeutic effect of ZST against DpA loss in PD mice since the treatment with in situ photoelectric stimulation markedly reversed the downregulation of TH, with increased expression of c-fos, compared with both the normal control group and the MPTP treatment group.

## Discussion

Here, we built a nanoscale photoelectrode with a zinc porphyrin self-assembled nanocrystal as the core and TiO$_2$ coated on the outside for high spatiotemporal modulation of the neurons via a capacitive

mechanism. The rational organization of the porphyrin J-aggregates and TiO$_2$ in the photocapacitive nanostructures was important for the enhancement of light harvesting and photoelectron transfer, thus yielding high photocapacitive efficiency. The positive and negative photoelectric potentials generated at the inner ZS and the outer TiO$_2$ drove the electrolytic cations around the cell membrane toward the electron-rich TiO$_2$; this resulted in membrane depolarization and AP generation. Notably, ZST could be alternatively excited by the two-photon process at the near-infrared bandwidth, with the potential for deep tissue neuron modulations. Compared with other photoelectrode neural stimulation nanomaterials, our photoelectrodes had

**Fig. 6 | The therapeutic mechanism of Parkinson's Disease in mice. a** High-frequency photoelectric stimulation of ZST directly activates STN efferent axons or somata, which is propagated to the internal segment of the pallidum (GPi) for improved PD's motor symptoms, as well as the substantia nigra pars compacta (SNc) for enhanced dopaminergic neurons' activity to further prevent the hyper-activity and abnormal rhythmic burst firing in STN (Created with BioRender.com.). **b** Photomicrographs of a representative mice brain injected with ZST in STN. **c** Representative traces of sEPSCs recorded in brain slices from the STN of mice in Con, PD, PD + ZST, PD+Laser, and PD + ZST+Laser. **d** Identical sEPSC frequencies and amplitudes. ($N = 9$ neurons from 3 mice). **e** Representative traces of sIPSCs recorded in brain slices from the STN of mice in Con, PD, PD + ZST, PD+Laser, and PD + ZST+Laser. **f** Statistical analysis of sIPSC frequencies and amplitudes indicating significant increase in the sIPSC frequency. ($N = 9$ neurons from 3 mice). **g** Immunofluorescent staining of TH and c-fos expression in SNc collected 2 hours after the last treatment (as the Flow chart of the treatment shown in Fig. 5d). **h** The statistical numbers of DpA neurons in different treatment groups. ($N = 10$ each group). **i, j** Western blotting and statistical analysis of TH and c-fos in control and MPTP-treated PD group. ($N = 3$ mice in each group). **k, l** Western blotting and statistical analysis of TH and c-fos in PD, PD + ZST, PD+Laser, and PD + ZST+Laser. ($N = 3$ mice in each group). The results are shown as mean ± SEM by one-way ANOVA analysis followed by Bonferroni's post hoc test for Con and PD. The results are shown as mean ± SEM by two-way ANOVA analysis followed by Bonferroni's post hoc test for PD, PD + ZST, PD+Laser, and PD + ZST+Laser. Data are representative of at least three independent experiments with similar results.

some particular features, such as adjustable nanosizes suitable for in vivo injection with high-spatial resolution, a broad excitation light wavelength range from visible to NIR for both in vitro and in vivo neuromodulation, and low work power density to ensure high bio-safety (Supplementary Dataset 2). This approach did not require gene transfer and was much simpler for neural stimulation at low laser power density than the current optogenetic techniques (Supplementary Dataset 3). Notably, ZST could be alternatively excited by the two-photon process at near infrared bandwidth, with the potential for deep tissue neuron modulations. Nanoscale photoelectrode devices could be developed as injectable prostheses for the treatment of neurological disorders, stroke, spinal cord injury, and even vision restoration.

## Methods

### Materials and Reagents
Zinc 5,10,15,20-tetrapyridylporphyrin (ZnTPyP) was purchased from J&K Scientific LTD. 4% Paraformaldehyde was purchased from Beyotime. MPTP (1-methyl-4-fenyl-1,2,3,6-tetrahydropyridin) (Adamas life, Y39211C) was purchased from Adamas Life. Sodium dodecyl sulfate (SDS), sodium hydroxide (NaOH), hydrochloric acid (HCl, 36%-38%), methanol, dimethyl sulfoxide (DMSO), acetonitrile and ethanol were purchased from Sinopharm Chemical Reagent Co., Ltd. (Beijing, China). 2.5% glutaraldehyde, Poly-l-lysine (PLL), 5,5-dimethyl-pyrroline-N-oxide (DMPO), 5,5'-Dithiobis-(2-nitrobenzoic acid) (DTNB), and titanium diisopropoxidebis (acetylacetonate) (TDAA), Glutathione, K-Gluconic acid, HEPES, Sodium Ascorbate, Thiourea, Sodium Pyruvate, N-acetyl-L-cysteine, $MgSO_4 \cdot 7H_2O$, Glucose, Cs-Methanesulfonate, TEA-Cl, $MgCl_2$, EGTA, Mg-ATP and Na-GTP were purchased from Sigma-Aldrich. All reagents were of analytical grade and were used without further purification.

### Animals
C57BL/6 J male mice (8–10 weeks) and 16-day Shjh:SD pregnant rats were purchased from Shanghai Jihui Laboratory Animal Care Co. All the experiments were bred separately and housed in the specific pathogen-free (SPF) animal facilities in the East China Normal University (light/dark cycle 10 h:14 h, temperature 20–26 °C, humidity 40–70%). All mouse and rat experiments were approved by the East China Normal University Animal Care and Use Committee. The Tab of Animal Experimental Ethical Inspection No. is m⁺R20190701. All the surgeries were performed under a general anesthetic condition to minimize suffering. All the mice used in the experimental groups were randomly assigned.

### Characterization
Nanocomposite particles were visualized using a 200 CX transmission electron microscope operated at 200 kV (JEOL Ltd., Tokyo, Japan). Scanning electron micrographs were obtained on a field emission JSM-6700F microscope (JEOL Ltd.). Luminescence emission spectra were collected on the fluorescence spectrometer (Edinburgh instruments, FLS 980). The element concentrations of samples were measured by inductively coupled plasma optical emission spectrometry (ICP-OES, Agilent Technologies 5100). The UV/Vis absorption spectrum was obtained using a UV-visible-NIR spectrophotometer (Shimadzu). The dynamic light scattering (DLS) was measured on nanoparticle size analyzer (Microtrac, nanotrac wave II). Two-photon properties were tested on a special ultrafast laser testing system equipped with fs laser (Chameleon Ultra II, 80 MHz, 140 fs). A mode-locked femtosecond Ti: Sapphire laser (MaiTai, Deepsee 80 MHz) on Nikon multi-photon A1R MP System was used for in vitro and in vivo light stimulation. X-ray diffraction characteristics were obtained using Bruker D8 Advance (Germany). UV-vis diffuser reflectance spectrum (DRS) was obtained using Hitachi U-3900H with $BaSO_4$ as a white standard. Mott-Schottky plots were carried out on an electrochemical workstation with a three-electrode system. The as-prepared materials spread on glassy carbon as working electrode, and Ag/AgCl electrode (3 M KCl) and Pt plate were used as reference and counter electrodes, respectively. 0.5 mol/L $Na_2SO_4$ solution was used as electrolyte deoxygenated using an $N_2$ stream. The photocurrent responses were measured on the electrochemical system mentioned above in 0.2 M $Na_2SO_4$ electrolyte using 300 W Xenon lamp under visible light radiation (400–800 nm).

### Synthesis of ZS and ZST
To synthesize ZS, SDS was dissolved in 9.5 mL NaOH solution ([SDS] 2 mM, [NaOH] 10 mM), and then 0.5 mL of ZnTPyP solution (0.01 M ZnTPyP dissolved in 0.2 M HCl) was quickly injected into the above solution with mild stirring for 24 h. Then, the mixture was subsequently centrifuged at 11872 g and washed with ethanol three times to remove surfactant, and the final ZSs were dissolved in ethanol. To coat $TiO_2$ on ZS, TDAA methanol solution ($V_{TDAA}$: $V_{MeOH}$ = 1:50) was cautiously added (interval time: 30 min, volume: 5 μL, 15 μL, 25 μL, 40 μL at a time, two times) into 8 mL ZSs. After the introduction of the titanium source, the reaction was continued for another 12 hours, and the final uniform forming of core-shell ZST was collected by centrifugation.

### Imaging of exciton migration
488 laser was introduced into microscope (Ti2-E, Nikon) by the commercial FRAP module (NIKON), reflected by the high-flatness filter set (Excitation: 483–494 nm, Dichroic Mirror: 497–553 nm, Emission: 581-625 nm), passed through high NA objective (SR HP ×100 Oil, NA 1.49, Nikon) and finally focused into one laser spot on the sample chamber. To obtain the size of the laser spot, one blank region (without any nanorod) was selected and illuminated with very high laser power, and due to incomplete blocking of DM, a small part of the reflected light from the sample still could reach CMOS target surface (Flash 4.0, Hamamastu). According to the image, size of the focused spot (Full Width of Half Maximum, FWHM) could be measured smoothly. Under the unchanged light path, one ZS microrod was moved to the position of the illuminated laser spot, and corresponding fluorescence images were captured, which were compared with reflection images to judge the existence of exciton migration.

## Two-photon properties of ZST

The transmittance of a nonlinear medium through a finite aperture as a function of the sample position z was measured with respect to the focal plane on a special ultrafast laser testing system equipped with fs laser (Chameleon Ultra II, 80 MHz, 140 fs). The two-photon absorption cross-section σ was calculated according to the reported method[35]. The σ was calculated by the following formula (1):

$$T(Z) = 1 - \frac{\beta I_0 L}{2^{\frac{3}{2}}(1 + (\frac{Z}{Z_0})^2)} \quad \sigma = \frac{\beta h\nu^* 10^3}{Nc} \tag{1}$$

$I_0$: light intensity at the focus; $L$: the thickness of sample; $N$: Avogadro constant; $c$: the concentration of solution; $\beta$: the nonlinear absorption coefficient.

In our experiment, the excitation light is 850 nm fs laser, the lens focus length is 3 cm, the thickness of sample is 1 mm, and the power used during the experiment was recorded.

## Ultrafast transient absorption (TA) spectroscopy

The TA data were recorded on a femtosecond pump-probe spectrometer coupled to an ultrafast amplified laser system. The fundamental beam was generated in a Ti/sapphire laser system (Astrella, 800 nm, 100 fs, 7 mJ/pulse, and 1 kHz repetition rate, Coherent Inc.), and divided into two portions by a beam splitter. One part is used to pump an optical parametric amplifier (OPerA Solo, Coherent Inc.), and the OPA can generate pulses with the needed wavelength to be used as the pump beam. The other part of 800 nm is used for the white-light generation, which is used as the probe beam. A femtosecond to microsecond transient absorption spectrometer (Helios Fire, Ultrafast System) was used to measure TA signals. A femtosecond to microsecond transient absorption spectrometer (Helios Fire, Ultrafast System) was used to measure TA signals. All of the investigated samples were well dispersed in ethanol, and during the measurements, the sample in the cell was constantly stirred under ambient conditions.

## Exciton binding energy of Zinc porphyrin

The optimization of the Zinc porphyrin's geometry was conducted within the framework of DFT with PBE0 functional[36] and def2-SVP basis set[37]. In the calculation, we also incorporated the DFT-D3 dispersion correction method along with the SMD (Solvation Model Based on Density) implicit solvent model[38]. All these DFT calculations were performed with Gaussian 16 program suite. To explore the photophysical properties, we conducted calculations on the excited electronic structure of the molecule using the time-dependent density functional theory (TDDFT) method at the PBE0-D3/def2-SVP level. Additionally, the exciton binding energy of the $S_1$ state was determined through an analysis of the TDDFT results using the Multiwfn program[39]. Visualization of the frontier molecular orbitals was generated using the Visual Molecular Dynamic program[40].

The exciton binding energy was calculated from the formula:

$$E_b = E_g - E_{opt} \tag{2}$$

where $E_{opt}$ and $E_g$ is optical and fundamental gap, respectively.

## Photothermal performance

The wavelength of 532/670 nm at power density of 23.5 and 41.6 mW/cm² was used as a light source to irradiate the ZST (50 µg/mL, 1 mL) solution dispersed in PBS. The average temperature of the solution was continuously monitored by infrared thermal imager (FLIR Tools) for 5 minutes of irradiation.

## GSH consumption for dye regeneration

DTNB was used to detect the GSH consumption as reductive scavengers for dye regeneration. The absorbance at 412 nm was measured by adding equal volume of PBS and DTNB in previously treated solutions by Laser, ZST, or ZST+Laser. ZST concentration was 100 µM; GSH solution concentration was 10 µM; DTNB solution concentration was 20 µM. 532 nm laser irradiated for 3 min.

## O₂·⁻ detection in solution

EPR spectra of solution with DMPO as radical scavenger, with or without ZST, and with or without laser irradiation (532 nm laser, power density 13.5 mW/cm²) were detected on Bruker EMX EPR spectrometer. Superoxide anion kit (Total Superoxide Dismutase Assay Kit with NBT) was used for quantitative detection of $O_2^{\cdot-}$ with varied ZST concentrations after 532 nm laser irradiation for 3 min.

## In vitro cytotoxicity assessment

Cell Proliferation and Cytotoxicity Assay Ki (Beyotime MTT Cell Proliferation and Cytotoxicity Assay Kit) was used for In vitro cytotoxicity assessment. Cells were seeded into a 96-well plate at $10^4$/well and then cultured at 37 °C for 24 h. Typically, different concentrations of ZST in the culture media were added into the wells and co-incubated for 24 h. For light treatment, 532 nm laser directly illuminated upon the 96-well plate. Then the treated cells continued to culture for another 12 h, followed by the addition of 100 µL MTT dissolved in PBS (0.6 mg/mL) for another 4 h. The absorbance of each well was monitored after the addition of DMSO by a microplate reader at the wavelength of 490 nm. The cell cytotoxicity was finally expressed as the percentage of cell viability in contrast to untreated control cells.

## Primary culture of rat cortical neurons

Pregnant rats at 18 days of gestation were anesthetized with halothane, and the fetuses were extracted. Then the brains of fetuses were removed rapidly and placed in ice-cold PBS. Tissues were dissected and incubated with 0.05% trypsine-EDTA (Gibco,15050065) for 15 min at 37 °C, followed by trituration with fire-polished pipettes, and plated in poly-L-lysine-coated culture dishes. Neurons were cultured with Neurobasal medium (Gibco) supplemented with B27 (Gibco, 17504-044) and maintained at 37 °C in a humidified 5% CO₂ incubator. Cultures were changed twice a week and used for all the assays 14–16 days after plating. A total of 12 pregnant rats were used for all cell experiments.

## Scanning electron microscopy images

The rat cortical neurons were cultured to maturity on a glass coverslip. ZST was added to the culture medium incubated with neurons for 12 h, and then cleaned with PBS. Next, the sample was then treated with 2.5% glutaraldehyde and stained at room temperature with 1% osmium tetroxide for 1 h. The sample was then dehydrated with a gradient concentration of ethanol and dried at the critical point. Images were taken on Zeiss GeminiSEM450.

## Confocal fluorescence imaging

ZST was added to the rat primary neurons in culture dishes. After co-incubation for 12 h, the cells were washed three times with PBS to remove the free nanoparticles. The luminescence imaging experiments were carried, showing that ZST nanoparticles were attached on the extracellular surface. For intracellular Ca²⁺ imaging, cortical neurons were co-cultured with Cal-520, AM (0.1 M, 4 µL, AAT Bioquest) for 2 h, and then washed with PBS for three times. The synchronous 405 nm laser stimulation and visible laser imaging at 488 nm was used to real-time show the changes of Ca²⁺ fluorescence intensity. The power density of 405 nm laser in confocal instrument (Nikon A1 + R-980) is about $10^3$ W/cm², and the laser is fixed as a continuous light source in the standard confocal microscope

## In vitro neuron electrophysiology experiments

Brain neurons were patch-clamped in the whole-cell current-clamp configuration using an Axopatch 700B amplifier (Molecular Devices). Cells in the culture were co-cultured with ZST for 12 h to ensure their effective attachment to neurons. The cells were first washed with our configured bath solution (NaCl 132 mM, KCl 4 mM, $MgCl_2$ 1.2 mM, $CaCl_2$ 1.8 mM, HEPES 10 mM, glucose 5.5 mM, pH 7.4) twice to remove the free particles, and then cultured in this bath solution. Borosilicate glass pipettes pulled on a $CO_2$ laser micropipette puller (Sutter Instruments P-2000) produced 4 MΩ resistances when filled with internal pipette solution (K-Gluconic acid 110 mM, NaCl 10 mM, $MgCl_2$ 1 mM, EGTA 10 mM, HEPES 30 mM, Mg-ATP 3 mM, Na-GTP 0.3 mM, pH 7.2). Under the current-clamp configuration, the triggered action potentials were recorded upon the illumination of 532 nm laser.

## Secondary motor cortex stimulation in vivo

Mice were anesthetized by i.p. injection of urethane (Sigma, St. Louis, MO, USA; 2 g/kg), mice were fixated in a stereotaxic apparatus (RWD Life Science, Shenzhen, China) for surgery. The degree of anesthesia was verified via the toe pinch method before the procedure started. To keep the mice's warm, a homeothermic blanket was set to 37 °C and placed under the anesthetized mouse. Erythromycin was then applied to both eyes of the mice throughout the experiment. Hair removal cream was used to remove the head, back and both hind limbs. The hair over the mouse back and hind limbs was removed to allow the movement the hind limbs movement. Then mice were pre-injected 1.5 µl ZST (50 µg/mL) in secondary motor cortex. The stereotaxic coordinates of the secondary motor cortex are anteroposterior (AP) +1.0 mm, mediolateral (ML) +0.5 mm, dorsoventral (DV) −0.5 mm. 570 nm one-photon or 850 nm two-photon laser was applied to illuminate the secondary motor cortex, while a video camera was used to capture the motions of both of the mouse's hind limbs during laser stimulation. Limb displacement was analyzed by computing the maximum displacement of the marker (green dot).

## In vivo electrophysiological recording

Nickel-titanium microwire electrodes (KD-MWA-F, KedouBC) were used for in vivo electrophysiological recording after ZST was injected in anesthetized mice. Four screws was implanted into the surface of the skull as the grounding and reference electrode. After implantation of the grounding screw, electrodes were slowly implanted into the right motor cortex (Anteroposterior (AP): anteroposterior (AP) +1.0 mm, mediolateral (ML) +0.5 mm, dorsoventral (DV) −0.5 mm, after reaching the targeted regions, and then the electrodes were glued to the skull using LELE® dental cement. The electrodes were connected to Zeus high-throughput neural recording system (Bio-Signal Technologies), and the 0–80 set screw was used as a reference. Electrophysiological recordings were acquired with a 20 kHz sampling rate and a 60 Hz notch filter. The signals were then analyzed using Offline Sorter ×64 V4 and NeuroExplorer 5 ×64. A total of seven neurons collected from 3 animals were used for statistical analysis.

## Treatment of MPTP-induced PD C57BL/6 J mice

Nine-week-old male C57BL/6 J mice weighing 20–30 g were divided into 5 groups, each consisting 8 mice. The groups were control (i.p. 0.9% saline), MPTP (i.p. 20 mg/kg), MPTP (i.p. 20 mg/kg) + ZST (1 µL), MPTP (i.p. 20 mg/kg) + laser, MPTP (i.p. 20 mg/kg) + ZST (1 µL) + laser. MPTP (Adamas life, Y39211C) was continuously given for 7 days. ZST was injected into the STN (anteroposterior (AP) −2.06 mm, mediolateral (ML) +1.5 mm, dorsoventral (DV) −4.5 mm) at a rate of 50 nL/min through a Gaoge syringe, waiting 5 min after injecting every 500 nL, and the needle remained in brain for 10 minutes after the injection was complete. Then an optical fiber was placed into the STN and cemented with dental acrylic. Mice were then single-housed for at least five days for recovery from lesions and surgery before the

behavioral paradigms. Mice were connected to a 532 nm laser via fiber optic cables and placed inside a testing chamber. For fixed 130 pulses per second optical stimulation, pulses were delivered in trains of six trials (each trial was composed of 10 s on and 20 s off) for a total of 3 min each day. After 10 days of light stimulation, the immunostaining and western blot of TH and c-fos expression in SNc collected 2 hours after the last treatment, slice electrophysiology and behavioral experiments were performed.

## Open-field test

We utilized an open-field system to assess spontaneous locomotor activity. Mice were acclimated in the experimental room for ~1 hour before being placed directly into the center of a 50 cm × 50 cm open field. Their movements were recorded via video during a 5-minute testing session. Subsequently, the animals were returned to their home cages. To eliminate olfactory cues, the open-field area was cleaned with ethanol after each measurement. The assessment metric involves the distance covered within the central nine squares, while overall activity distance is determined by the path crossing all squares during the test.

## Gait analysis

Gait analysis was performed using the Catwalk™ XT (Noldus Information Technology, Wageningen, The Netherlands), and the parameters were then calculated using the Catwalk XT software package. Mice were trained on a Catwalk runway to perform some uninterrupted runs 1 day prior to surgery to relieve stress. The actual test was repeated until three consecutive uninterrupted runs were recorded. Mice that were visually observed to exhibit prolonged stopping or turn backward on the runway were considered to fail the test.

## Slice electrophysiology

A vibratome (Leica VT1000S) was used to cut STN-containing brain slices (thickness: 350 µm) in ice-cold cutting solution, which contains: 92 mM NMDG, 93 mM HCl, 2.5 mM KCl, 1.2 mM $NaH_2PO_4 \cdot 2H_2O$, 30 mM $NaHCO_3$, 20 mM HEPES, 5 mM Sodium Ascorbate, 2 mM Thiourea, 3 mM Sodium Pyruvate, 12 mM N-acetyl-L-cysteine, 10 mM $MgSO_4 \cdot 7H_2O$, 25 mM Glucose and 0.5 mM $CaCl_2 \cdot 2H_2O$. The brain slices were incubated for 45 min at room temperature in artificial cerebrospinal fluid (aCSF) solution containing: 92 mM NaCl, 2.5 mM KCl, 1.2 mM $NaH_2PO_4 \cdot 2H_2O$, 30 mM $NaHCO_3$, 20 mM HEPES, 5 mM Sodium Ascorbate, 2 mM Thiourea, 3 mM Sodium Pyruvate, 2 mM $MgSO_4 \cdot 7H_2O$ 25 mM Glucose, 2 mM $CaCl_2 \cdot 2H_2O$. The recording aCSF contained: 140 mM NaCl, 4.7 mM KCl, 2.5 mM $CaCl_2$, 1.2 mM $MgCl_2$, 11 mM D-glucose, and 10 mM HEPES (pH 7.2). All solutions were bubbled with 95% $O_2$ and 5% $CO_2$ throughout the experiment. MultiClamp 700B amplifier (Molecular Devices, USA) was used for whole-cell recordings of sIPSC and sEPSC; aCSF was supplemented with 50 µM APV (sigma) and 20 µM CNQX (sigma), or 100 µM bicuculline (sigma), with voltage holding at 0 mV and −60 mV, respectively. Pipettes were filled with an internal solution containing 100 mM Cs-Methanesulfonate, 10 mM NaCl, 10 mM TEA-Cl, 30 mM HEPES, 1 mM $MgCl_2$, 10 mM EGTA, 3 mM Mg-ATP, and 0.3 mM Na-GTP. Using pClamp10 (Molecular Devices, USA), data were filtered during acquisition with a low-pass filter of 2 kHz, and analyzed offline with Clampfit 10.4 (Molecular Devices, USA).

## Western blot

The brain tissue was homogenized using a tissue grinder (Kimble) in 1.5 ml tubes containing lysis buffer composed of 50 mM Tris pH 7.5, 150 mM NaCl, 5 mM EDTA pH 8.0, 1% SDS, and a protease inhibitor cocktail. Subsequently, the homogenates were centrifuged at 14,000 × $g$ for 10 minutes at 4 °C. The supernatant was collected and heated in a dry bath at 75 °C for 20 minutes to denature proteins. The denatured tissue lysates were then resolved using SDS-PAGE (Bio-Rad)

and transferred onto nitrocellulose membranes. Following a 1-hour block in Tris-buffered saline with 5% non-fat milk and 0.5% Tween-20 (TBST), membranes were incubated overnight at 4 °C with primary antibodies. The next day, after three washes with TBST, membranes were incubated at room temperature for 1 hour with HRP-conjugated secondary antibodies (CWS) in TBST containing 5% milk powder. Subsequently, membranes were washed three times with TBST. Bands were visualized using the Immobilon Western ECL system (Bio-Rad, USA), and data analysis was performed using Gel-Pro Analysis (Media Cybernetics, USA). Anti-β-actin antibody (Abcam, ab8227, 1:1000), anti-c-Fos antibody (Abcam, ab214672, 1:500), anti-Tyrosine Hydroxylase antibody (Abcam, ab6211, 1:500) were used. All the uncropped blots were supplied in the Source data file.

### Histology study

After the above experiments, mice were deeply anesthetized with Pelltobarbitalum Natricum (40 mg/kg, i.p.) and perfused transcardially with 0.1 M PBS followed by 4% paraformaldehyde in 0.1 M PBS. Brains were postfixed in 4% paraformaldehyde overnight at 4 °C and then transferred to 30% sucrose (4 °C). The brains were cut into 30-μm sections in the coronal plane using a cryostat and processed for staining. TH immunofluorescence staining was used to determine the extent of degeneration of dopaminergic neurons. The c-fos immuno-histochemistry was used to detect the change of c-fos protein expression in brain. The GFAP and Iba1 immunohistochemistry were used to suggest the biocompatibility of ZST. Anti-c-Fos antibody (Abcam, ab214672, 1:500), anti-Tyrosine Hydroxylase antibody (Abcam, ab6211, 1:500), anti-GFAP antibody (Abcam, ab68428, 1:500), anti-IBA1 antibody (Wako, 019-19741, 1:500), and anti-NeuN antibody (Abcam, ab236869, 1:500) were used.

### Immunofluorescence of cultured neurons

Cultured neurons were washed 3 times, fixed with 4% paraformaldehyde for 10 minutes, treated with 0.25% TritonX-100 (sigma) for 10 min, blocked with 5% bovine serum albumin, and then incubated overnight with GFAP, NeuN antibody at 4°C. After being washed for 3 times, cells were incubated with Alexa488-conjugated or F594-conjugated secondary antibodies. Thereafter, cells were washed and stained with Hoechst (Beyotime, P0133), and finally analyzed by laser confocal microscope. Finally, neurons were observed by laser confocal microscope and analyzed by ImageJ software. Anti-GFAP antibody (CST, 3670 S, 1:500), anti-NeuN antibody (Abcam, ab236869, 1:500) were used.

### Statistical analysis

We conducted statistical comparisons between two groups using one-way analysis of variance (ANOVA) and Bonferroni post hoc analyses. The two-way repeated measures ANOVA, two-way ANOVA, and Bonferroni post hoc analyses were used in analyses with multiple experimental groups. Data are shown as individual values or expressed as the mean ± SEM, and significance levels are indicated as $*p < 0.05$, $**p < 0.01$, $***p < 0.001$, and not significant (n.s.).

### Reporting summary

Further information on research design is available in the Nature Portfolio Reporting Summary linked to this article.

## Data availability

The main data supporting the results in this study are available within the paper and its Supplementary Information. All data generated in this study are available from the corresponding authors. Source data are provided with this paper. The source data of linear graph and column diagram and uncropped scans of Western blots in the figures are provided as a Source Data file. Source data are provided with this paper.

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

## Acknowledgements

The authors would greatly acknowledge the financial support by the Key Program of National Natural Science Foundation of China (Grant No. 22235004 to W.B.B.), Innovation Program of Shanghai Municipal Education Commission (No. 2023ZKZD01 to W.B.B.), National Natural Science Foundation of China Youth Fund (Grant No. 52322213 to Y.Y.L.), the National Funds for General Projects (No. 52272269 to Y.Y.L.), Shanghai Rising-Star Program (No. 21QA1400900 to Y.Y.L.), Natural Science Foundation of Shanghai (No. 21ZR1405300 to Y.Y.L.). Thanks to Qing-feng Xiao in NIKON INSTRUMENTS (SHANGHAI) CO., LTD. for vigorous support in optical design and imaging.

## Author contributions

W.B.B., Y.Y.L., Y.M., and D.Y.J., conceived the idea and designed the study; Y.Y.L. performed the material experiments and data analysis; X.L.W. and J.Q.C. performed the transient absorption spectra and data analysis; J.C., and F.X.C., P. P. F. cultured the in vitro neurons and performed cellular experiments; J.C. and M.N.G. performed the in vivo electrophysiological recording and data analysis; J.C. and F.X.C. performed the electrophysiological experiments both in vivo and in vitro; J.J.Z., D.Y.J., W.H., X.Z.H., W.W.H. and J.H. contributed to discussion; W.B.B., Y.M., and D.Y.J. supervised the research. All the authors participated in the preparation of the manuscript.

## Competing interests

The authors declare no competing interests.
