## [Peer Review File · Nature Communications]

REVIEWER COMMENTS

Reviewer #1 (Remarks to the Author):

This work reports intriguing results of in vitro and in vivo light stimulation of cell activity using a new transducer composed of a molecular aggregate, claimed to be a J-aggregate, coated with a TiO₂ shell. While this approach is original, it is worth noting that there are existing examples in the literature involving similar techniques, such as silicon nanorods, carbon nanotubes, inorganic quantum dots (QDs), and organic polymer nanoparticles. Regrettably, these existing studies have not been adequately referenced. It is advisable to include relevant publications in the bibliography and engage in a comparative discussion of these approaches with the one proposed here.

The extensive characterization of the transducers is commendable. However, the overall picture obtained from the results may require further validation for full persuasiveness. A number of statements within the paper appear to be speculative and diverge from commonly accepted knowledge. Additionally, the implications of the proposed model are not thoroughly discussed.

In its current form, the paper exhibits several flaws. The language can be cumbersome and at times challenging to comprehend. Consequently, I am unable to recommend its publication.

In more detail, here are a few observations.

Several excitation wavelengths are used, from 400 nm to 850 nm, but there is no clear rationale on why one or the other, except for the two-photon stimulation. For instance, why at some point it is 532 nm, later 670 nm and so on?

The absorption spectra of the J aggregate should be fully explained. In particular all the bands need to be assigned, at least tentatively. Upon aggregation the "J" band is very broad, indeed broader than that of the monomer, this is opposite what usually occurs. In addition, there are other absorption peaks around 600 nm that are not assigned nor discussed.

Same is true for the PL spectrum, why did two bands appear there?

I think this statement: "...where the hot carriers flow into the conduction band of TiO₂ for photo generated exciton separation and subsequent TiO₂(e⁻) formation..." need deep clarification. Authors claim the exciton splits into "hot" charge carriers, that eventually reach TiO₂. What is the driving force leading to exciton ionization? What is the exciton binding energy and the yield of ionization? Can authors expose evidences for the very existence of such carriers inside the J-aggregate?

Tauc's plot is suitable for band-like crystalline semiconductors, not the case here. In addition, authors talk about a band gap, is this an electrical band gap associated to charge generation or an optical band gap associated to exciton formation?

Regarding transient absorption spectroscopy data, I presume the signal is positive (bleaching)? This should be explained, and the color scale should have associated scale-labels. In the text authors write: line 157-158 “..From the kinetic traces at 540 nm, electron injection was observed in ZST occurring much faster ($\tau_0 = 370 \pm 80$ fs) than that of ZS ($\tau_0 = 840 \pm 60$ fs),..”. Why there should be an electron injection also in ZS, where electrons are injected?

The hole recombination leads apparently to the generation of super oxide that is free to reach the biological tissue- Superoxyde is a powerful ROS that can lead to several toxic reactions. Authors should explain which is the fate of this superoxide and why, in their opinion, it is not toxic.

In vitro measurements seem to show that the rods fully cover the neuron surface. In these conditions thermal effect might have a role to play, and it should be considered before being ruled out.

I.42 “..higher stimulation voltage” is comparative with what?

I.44 “reaction oxygen generation” is a mistyping

I.52 the solution proposed here do need as well optical fibers to reach deep brain regions. This should be fully acknowledged and discussed.

I.66 “surface, achieving direct charges injection in electrolyte via a capacitive process”. A capacitive process does not inject charge carriers, by definition.

I.79 “single-photon” is misleading, it should be one-photon

Minor comments:

1. L. 79 Non-Linear Optics is usually abbreviated as NLO, not “NOL”.

2. L. 112 “across section” is cross-section.

Viability: (i) could the authors explain the choice to use MTT assay for testing the viability of neurons instead of FDA/PI staining? (ii) histograms reported in the Extended Figure S5 present differences across the experimental conditions. Did the authors perform statistical analysis on those results? Are those differences statistically and/or biologically significant?

Figure 2: Labels 1 and 2 are swapped between text and plot.

Figure 3: (i) it would be clearer for the readers including a graph of the percentage of failure (of AP) inside the main figure. (ii) please change the colors of the conditions 0.5 and 2.0ms (too similar).

Figure 4: (i) explain the choice not to investigate microglia besides astrocytes.

(ii) authors should provide information of the morphology of the astrocytes (not only the number) as an additional indication of the absence of astrogliosis.

Figure 6: is the increase in the sIPSC frequency related to a modification of the number of synaptic boutons and/or Ca²⁺ dynamics? Please comment on this and/or provide new data.

Materials and methods section:

(i) authors did not report the level of purity of the neuronal culture. Are astrocytes present in primary cultured neurons? any idea about the percentage?

(ii) why the authors chose to use a pipette solution with high concentration of Cl⁻?

(iii) why the authors avoided to addition of CGP (or other specific blocker of GABAB receptor) for the sPSC recordings? Please comment on this.

(iv) it lacks a paragraph explaining the statistical analysis.

Reviewer #2 (Remarks to the Author):

The manuscript describes the design and validation of nanoparticles for neuromodulation using photoelectric materials. It details the synthesis of nanoparticles of ZS which are then coated with TiO₂ to produce ZST in the 100s of nm size range. Characterization of the nanoparticles in terms of optical properties validates its sensitivity to the given wavelengths and their efficiency. ZST were then placed with cultured cortical neurons and demonstrated their ability to elicit action potentials from light, a fast effect with low energy density that was dependant on the nanorods. The particles were then injected into mice brains and activated with light to stimulate hindlimb activity without a distinct immune response. This was elicited with both single photon at 670 nm or two-photon at 850 nm. They were then used as a possible treatment for PD in mice. Overall, the manuscript is convincing for ZST as an effective acute light-based wireless neuromodulation device.

Major comments:

Localization of the ZST was validated after 30 days however their function was not addressed during this month. It begs the question of whether the nanorods were functional, and to what degree, during the 1-month chronic injection used to produce this histology. An assessment of at least short-term ageing in in vitro or in vivo conditions would address this concern.

The inclusion of the Parkinson's disease section is interesting however feels out of place given the main goal of the article is to present a novel technology for neuromodulation. Firstly, it requires the implantation of an optical fibre for use which negates the main benefit of ZST as untethered. Critically, the protocol presented does not seem to have been used in previous literature as a treatment so an equivalent electrical device may explain why this stimulation protocol (10 days of 3 minutes of pulse trains prior to behavioural tests) was chosen and how it was known that it would improve PD symptoms. In the given reference, stimulation is provided during the behavioural tests and not as a longer-term treatment. In this manuscript, it seems like the stimulation regrows neurons so that function is regained and is present even when the light stimulation is not continued. This is a very interesting result but needs more exploration. This section needs to be expanded to be convincing or should be incorporated in a separate article.

Minor comments:

This is a photocapacitive device, not photovoltaic.

Not all faradaic processes are irreversible. Reversible redox reactions are common in current neural devices.

Figure 1c is not clear in describing the working principle of the device as it appears charges enter the tissue, as opposed to charging a capacitor.

In Figure 3b, it is unclear on the location of the ZST. An outline of the nanorods would help.

For the acute hindlimb experiments, was the skin closed?

In the videos, it appears as though there is some residual movement when the light is off. With the 850 nm video, the red dot timing is sometimes off as the muscle twitches first.

Point-by-point responses to Reviewers

Reviewer #1 (Remarks to the Author):

This work reports intriguing results of *in vitro* and *in vivo* light stimulation of cell activity using a new transducer composed of a molecular aggregate, claimed to be a J-aggregate, coated with a TiO₂ shell. While this approach is original, it is worth noting that there are existing examples in the literature involving similar techniques, such as silicon nanorods, carbon nanotubes, inorganic quantum dots (QDs), and organic polymer nanoparticles. Regrettably, these existing studies have not been adequately referenced. It is advisable to include relevant publications in the bibliography and engage in a comparative discussion of these approaches with the one proposed here.

Response: Thanks for the reviewer's kind suggestion and we have added the literatures involving similar techniques, such as silicon nanorods, carbon nanotubes, inorganic quantum dots (QDs), and organic polymer nanoparticle. We categorized the nanomaterials used in nerve stimulation for a comprehensive comparison. This table also showed the advances presented in our work, with a particular focus on features such as adjustable nano-sizes suitable for *in vivo* injection with high spatial resolution, a broad excitation light wavelength range from visible to NIR for both *in vitro* and *in vivo* neuromodulation, and low work power density to ensure high biosafety.

Table S1. The comparison of different electrodes used in neural stimulation.

	size	Stimulus light: wavelength; power density	Invasiveness for in vivo study	Mode of stimulation	In vivo application	Reference
HgTe nanocolloid	10 nm	532 nm; 800 mW/cm ²	N/A	Capacitive	N/A	Nano Letters 7, 513-519, (2007).
P3HT: PEDOT:PSS	~μm	300 DPI; 20 cd m ⁻²	Chronic implantable in vivo	Uncertain	Restore vision	Nat. Mater. 16, 681–689 (2017).
PIN-Si nanowire	Diameter: ~200 nm Length: ~ μm	532 nm; 6-17 mW	N/A	Faradaic	N/A	Nat. Nanotechnol. 13, 260-266, (2018).

Silicon-based materials	Nanowire; membran, meshes	530 nm; 6 W cm ⁻²	Chronic implantable in vivo	Faradaic; Capacitive	Forelimb movement	Nat. Biomed. Eng. 2, 508-521, (2018).
Au-decorated TiO ₂ nanowire arrays	Diameter: 100 nm Length: 2 μm	375/28 nm, 470/20 nm, 546/12 nm; 470 μW/mm ²	Chronic implantable in vivo	Faradaic	Restore vision	Nat. Commun. 9, 786, (2018).
carbon nanotubes	50 nm ²	690 nm; 1 W	Chronic implantable in vivo	N/A	N/A	Ann Biomed Eng 38, 3500–3508 (2010).
InP/ZnS QF	3 nm	450 nm 169 mW/cm ²	N/A	Faradaic	N/A	Nano Letters 19, 5975-5981, (2019)
P3HT nanoparticles	300 nm	540 nm; 40 mW/mm ²	Minimally invasive injection	Capacitive	Restore vision	Nat. Nanotechno 1. 15, 698–708 (2020).
AlSb nanocrystals	9.1 nm	445 nm; 100 mW/cm ²	N/A	Capacitive	N/A	Commun. Mater. 2, 19 (2021).
Our materials: ZST	Varied length from ~100 nm to a few μm	400-700 nm; 530 nm laser; 850 nm fs laser; 20 mW/cm ²	No optical fiber stimulation in the motor cortex	Capacitive	Forelimb movement; Parkinson's Disease Treatment	Our work

The extensive characterization of the transducers is commendable. However, the overall picture obtained from the results may require further validation for full persuasiveness. A number of statements within the paper appear to be speculative and diverge from commonly accepted knowledge. Additionally, the implications of the proposed model are not thoroughly discussed.

Response: We appreciate the reviewer for the constructive suggestions. We have made the following enhancements to bolster the persuasiveness of the results presented in the article.

1. Supplementary references: We have added the relevant references of optoelectrodes used in neuromodulation researches, and made a table for a comprehensive parameter comparison and discussion to illustrate the innovation of our nanoelectrode-based neural modulation.

2. A full discussion of data: We have provided an exhaustive discussion of the experimental data, which encompasses spectral data, and material properties. Additionally, we have addressed language and expression errors, enhancing the overall quality of the article. A detailed attribution of peaks in the ultraviolet and fluorescence spectra was carried out to elucidate the optical properties of the material. The elucidation of the material's pertinent characteristics provides further evidence for experimental conclusions, primarily encompassing: (1) the source of driving force for exciton separation; (2) the reasons for the material's suitability for calculating bandgap using the Tauc plot; (3) the presence of potential toxicity of Superoxide.

3. Supplementary experiments: In the context of materials, we have computed the exciton dissociation efficiency (56%) and utilized Density Functional Theory (DFT) to determine the exciton binding energy of the material (0.70 eV) in order to further quantify the process of exciton separation in ZST. Experimental evidence through thermal imaging has demonstrated that ZST exhibits no significant thermal effects under illumination, with a temperature change of less than 1°C. Furthermore, we conducted immunofluorescence experiments to confirm the neuronal purity is up to $75.8\% \pm 2.4\%$. In both *in vivo* and *in vitro* experiments, we have also determined that ZST exhibits excellent biocompatibility. Notably, we found ZST could remain effective after being injected in brain tissue for 30 days through *in vivo* multi-channel experiments.

4. PD model mice: Since our optoelectrodes can activate the action potential of neurons upon ms pulsed light irradiation *in vitro*, we decided to test the effectiveness of our optoelectrodes *in vivo*. In clinical practice, high-frequency electrical stimulation is effective for Parkinson's disease (PD) treatment. So we constructed the PD model mice and applied the 130 Hz photoelectronic treatment for one week. We observed that this treatment can reduce the spontaneous firing activity of neurons in STN, thus preventing excessive activity and abnormal rhythmic burst firing. Furthermore, it has been observed this treatment increases the number of dopaminergic neurons' functional units, indicating that high-frequency stimulation might promote neurogenesis. All these results indirectly reflect the effectiveness of our optoelectrodes in neuromodulation *in vivo*, showing the potential to guide the exploration of the biological mechanisms in the treatment of neurological disorders.

In its current form, the paper exhibits several flaws. The language can be cumbersome and at times challenging to comprehend. Consequently, I am unable to recommend its publication.

Response: We appreciate this reviewer for the constructive suggestions. In the revised manuscript, we have made corrections to the errors and also polished the text to make it easier for readers to understand.

In more detail, here are a few observations.

Several excitation wavelengths are used, from 400 nm to 850 nm, but there is no clear rational on

why one or the other, except for the two-photon stimulation. For instance, why at some point it is 532 nm, later 670 nm and so on?

Response: We thank the reviewer for raising the question about the different lasers we used in different experiments. As shown in Fig 2c, since the J-aggregated porphyrin exhibits a broad absorption band compared with that of the porphyrin monomer, the excitation wavelengths from 400 nm to 700 nm all can excite our constructed optoelectrodes. The specific reasons for using different wavelengths of light are as follows:

1. The reason of using 405 nm in our *in vitro* calcium imaging experiments is that 405 nm is the only available stimulation laser for synchronous 405 nm laser stimulation and 488 nm calcium imaging in our confocal microscopy (Nikon A1). The use of a 405 nm light source is more convenient for the study of *in vitro* synchronous stimulation observation calcium imaging.
2. Considering that the max absorption peak of our optoelectrode was around 532 nm, we chose a 532 nm laser for *in vitro* and *in vivo* neuromodulation.
3. In addition, to be more conducive to the subsequent *in vivo* application, we conduct another set of experiments using 670 nm red laser with a better tissue penetration than 532 nm laser. Upon extracranial 670 nm laser irradiation, we confirm the ability of ZST to optically control the electronic activity of neurons *in vivo* without the need for optical.

The absorption spectra of the J aggregate should be fully explained. In particular, all the bands need to be assigned, at least tentatively. Upon aggregation the “J” band is very broad, indeed broader than that of the monomer, this is the opposite of what usually occur. In addition, there are other absorption peaks around 600 nm that are not assigned nor discussed.

Response: We thank this reviewer for the constructive suggestions.

1. The J aggregated ZST absorption spectrum is derived from transitions from the ground state (S_0) to the two lowest excited singlets S_1 (Q band) and S_2 (B band). The appearance of B band is caused by the $a_{1\mu}(\pi) - e_g(\pi^*)$ transition of electrons in the porphyrin ring, which is generally located between 400 and 450 nm. The Q band is produced by the $a_{2\mu}(\pi) - e_g(\pi^*)$ transition and is located around 600 nm^{1,2}.
2. In our work, the reason for the broadening of the peak of the porphyrin monomer after J aggregation is as follows: The peak of the porphyrin monomer in B band comes from the degenerate B_x and B_y transition, and after aggregation, it splits in B band, which leads to the broadening of the “J” band³. This broadening phenomenon also appears in the work of other porphyrin aggregations^{4,5}, so it may be the unique feature of porphyrin-based J aggregations that is different from other dye molecules.

3. The absorption peaks of ZST around 600 nm are Q (0,0) and Q (1,0) from the Q band. Q (0,0) refers to the excitation from the lowest vibrational level of the ground state singlet to the lowest vibrational level of the first excited singlet electronic state, and Q (1,0) has a quantum vibration in the first excited singlet electronic state⁶.

References:

1. Yang, S. I. et al. Interplay of orbital tuning and linker location in controlling electronic communication in porphyrin arrays. *J Am Chem Soc* 121, 4008-4018, doi:DOI 10.1021/ja9842060 (1999).
2. Bajju, G. D. et al. Synthesis and Spectroscopic and Biological Activities of Zn(II) Porphyrin with Oxygen Donors. *Bioinorg Chem Appl* 2014, doi:Artn 78276210.1155/2014/782762 (2014)
3. Wan, Y. et al. Exciton Level Structure and Dynamics in Tubular Porphyrin Aggregates. *J Phys Chem C* 118, 24854-24865, doi:10.1021/jp507435a (2014).
4. Cai, B. et al. Promoted Charge Separation and Long-Lived Charge-Separated State in Porphyrin-Viologen Dyad Nanoparticles. *J Am Chem Soc*, doi:10.1021/jacs.3c04372 (2023).
5. Okada, S. & Segawa, H. Substituent-control exciton in J-aggregates of protonated water-insoluble porphyrins. *J Am Chem Soc* 125, 2792-2796, doi:10.1021/ja017768j (2003).
6. Spellane, P. J., Gouterman, M., Antipas, A., Kim, S. & Liu, Y. C. Porphyrins .40. Electronic-Spectra and 4-Orbital Energies of Free-Base, Zinc, Copper, and Palladium "Tetrakis(Perfluorophenyl)Porphyrins. *Inorg Chem* 19, 386-391, doi:DOI 10.1021/ic50204a021 (1980).

Same is true for the PL spectrum, why did two bands appear there?

Response: We thank this reviewer for the constructive suggestions on the presence of two bands in the fluorescence spectrum. Since the energy level difference between the first excited singlet state (S_1) and the second excited singlet state (S_2) of porphyrin is small. Therefore, the direct transition of S_1-S_0 and S_2-S_0 occur, and two emission bands at 610 and 660 nm⁷.

References:

7. Bajju, G. D. et al. Synthesis and Spectroscopic and Biological Activities of Zn(II) Porphyrin with Oxygen Donors. *Bioinorganic Chemistry and Applications* 2014, 782762, doi:10.1155/2014/782762 (2014).

I think this statement: "...where the hot carries flow into the conduction band of TiO₂ for photo generated exciton separation and subsequent TiO₂(e-) formation..." need deep clarification. Authors claim the exciton splits into "hot" charge carriers, that eventually reach TiO₂. What is the

driving force leading to exciton ionization? What is the exciton binding energy and the yield of ionization? Can authors expose evidences for the very existence of such carriers inside the J-aggregate?

Response: Thanks for the rigorous thinking by this reviewer.

1. According to the research of dye-sensitized batteries, the driving force of exciton ionization comes from the energy level difference of the lowest occupied orbital of zinc porphyrins than the conduction band level of TiO₂^{8,9}.

2. The exciton binding energy (E_b) was calculated by DFT. The formula used is as follows: E_b=E_{fund}-E_{opt}, where E_{fund} is the base gap; E_{opt} is the light gap, defined as the difference between the ground state and the lowest excited state. The calculated value of E_b is 0.70 eV.

The yield of ionization is 56%. Based on the photo-activated excited state relaxation mechanism, the quantum yield of exciton splitting Φ can be estimated from the equations below:

$$\tau_{ZS-measurement} = \frac{1}{k_1} = 0.84 \times 10^{-12} s$$
$$\tau_{ZST-measurement} = \frac{1}{k_1 + k_2} = 0.37 \times 10^{-12} s$$
$$\Phi = \frac{k_2}{k_1 + k_2} = 1 - \frac{0.37}{0.84} = 0.56 = 56\%$$

3. As shown in Fig 2j, under excitation of 480 nm, the transient absorption spectra of ZS and ZST show changes. However, 480 nm laser couldn't excite TiO₂, but can only excite ZS, so the change of transient absorption spectrum can be used as evidence that the exciton of zinc porphyrin becomes a carrier.

References:

8. Calogero, G., Bartolotta, A., Di Marco, G., Di Carlo, A. & Bonaccorso, F. Vegetable-based dye-sensitized solar cells. *Chem Soc Rev* 44, 3244-3294, doi:10.1039/c4cs00309h (2015).

9. Muñoz-García, A. B. et al. Dye-sensitized solar cells strike back. *Chem Soc Rev* 50, 12450-12550, doi:10.1039/d0cs01336f (2021)

Tauc's plot is suitable for band-like crystalline semiconductors, not the case here. In addition, authors talk about a band gap, is this an electrical band gap associated to charge generation or an optical band gap associated to exciton formation?

Response: Thanks for the rigorous thinking by this reviewer. In our work, as shown in Fig S2a, ZS, and ZST formed a crystal structure. Therefore, we considered that it is suitable to determine the bandwidth through Tauc's plot¹⁰. And the result of Tauc's plot test was an optical band gap associated with exciton formation.

References:

10. Zhang, Y. N. et al. H₂O₂ generation from O₂ and H₂O on a near-infrared absorbing porphyrin supramolecular photocatalyst. *Nat Energy* 8, 361-371, doi:10.1038/s41560-023-01218-7 (2023).

Regarding transient absorption spectroscopy data, I presume the signal is positive (bleaching)? This should be explained, and the color scale should have associated scale-labes. In the text authors write: line 157-158 “.From the kinetic traces at 540 nm, electron injection was observed in ZST occurring much faster ($\tau_0 = 370 \pm 80$ fs) than that of ZS ($\tau_0 = 840 \pm 60$ fs),..”. Why there should be an electron injection also in ZS, where electrons are injected?

Response: Thanks for the reviewer's comments. We sincerely apologize for the omission of the color scale in the previous version of the manuscript. This error has been corrected. As shown in the Fig 2j, red color represents the positive signal, while blue color represents the negative signal. In transient absorption data, positive signals typically arise from excited-state absorption signals, while negative signals originate from ground-state bleaching signals. The kinetic curve detected at 540 nm shown in panel k is primarily associated with the singlet states absorption signal of ZnPy. For ZS, the fast component extracted from fitting the TA data with 840 ± 60 fs should respond to the lifetime of the S₂ state. The excited state population on S₂ state should mainly decay to the S₁ state by internal conversion. The lifetime shown in our work agrees with data reported earlier for other Zn(II) porphyrins^{11, 12, 13}. On the other hand, since the excited-state lifetime is influenced by all pathways, the lifetime of the S₂ state obtained in ZST is 320 fs, which is nearly three times faster than in ZS, suggesting that, following excitation to the S₂ state in ZST, there not only exists an internal conversion pathway but also a charge transfer pathway to TiO₂, i.e., an ultrafast charge injection process.

Proposed Photoactivated relaxation mechanism for ZS and ZST.

When a semiconductor such as ZS is photoexcited, due to the material's inherent surface state or dangling bonds, there are many bandgap states on its surface, which are called surface states or defect states. These states can trap the electrons generated by the photoexcitation, and similar electron injection processes also exist in other porphyrin-based assemblies¹⁴.

References:

11. Gurzadyan, G. G., Tran-Thi, T. H. & Gustavsson, T. Time-resolved fluorescence spectroscopy of high-lying electronic states of Zn-tetraphenylporphyrin. *J Chem Phys* 108, 385-388, doi:Doi 10.1063/1.475398 (1998)
12. Mataga, N., Shibata, Y., Chosrowjan, H., Yoshida, N. & Osuka, A. Internal conversion and vibronic relaxation from higher excited electronic state of porphyrins: Femtosecond fluorescence dynamics studies. *J Phys Chem B* 104, 4001-4004, doi:DOI 10.1021/jp9941256 (2000).
13. Gacka, E., Burdzinski, G., Marciniak, B., Kubas, A. & Lewandowska-Andralojc, A. Interaction of light with a non-covalent zinc porphyrin-graphene oxide nanohybrid. *Phys Chem Chem Phys* 22, 13456-13466, doi:10.1039/d0cp02545c (2020).
14. Wheeler, D. A. & Zhang, J. Z. Exciton Dynamics in Semiconductor Nanocrystals. *Adv Mater* 25, 2878-2896, doi:10.1002/adma.201300362 (2013).

The hole recombination leads apparently to the generation of super oxide that is free to reach the biological tissue- Superoxyde is a powerful ROS that can lead to several toxic reactions. Authors should explain which is the fate of this superoxide and why, in their opinion, it is not toxic.

Response: We thank this reviewer for the constructive suggestion on the fate and toxicity of this superoxide.

Superoxide, as an existing signal molecule, exists in cyclic equilibrium *in vivo*. The production of superoxide can be consumed by SOD enzymes that remove superoxide anions from the organism. There are three main forms of SOD enzyme: Cu/Zn SOD located in the cytoplasm, Mn-SOD located in the mitochondria, and EcSOD located in the cytoplasm. The main role is to remove superoxide

anions to produce H_2O_2 and O_2 , H_2O_2 under the action of catalase or reducing agents (GSH) to produce H_2O ¹⁵. Notably, the toxicity of superoxide anion is closely related to its content. The redox potential of the superoxide anion is about -0.33 eV, and the toxicity of the superoxide anion is much less than that of the hydroxyl radical, singlet oxygen, and other strong oxidizing free radicals in the cell as oxidant or reducing agent¹⁶. Moreover, superoxide anion is difficult to pass through the plasma membrane. Thus, the generated superoxide anion from ZST was mainly located outside neurons, which makes it difficult to affect the intracellular environment. In both cell and *in vivo* experiments, we evaluated the effects of materials and light on cell activity, no abnormalities were observed at the cellular level (e.g., normal neuron action potential release waveform) and tissue level (e.g., no inflammatory response) after stimulation. The results showed that the materials had good biosafety under light compared with the control group.

References:

15. Borgstahl, G. E. O. & Oberley-Deegan, R. E. Superoxide Dismutases (SODs) and SOD Mimetics. *Antioxidants-Basel* 7, doi:ARTN 15610.3390/antiox7110156 (2018).

16. Nosaka, Y. & Nosaka, A. Y. Generation and Detection of Reactive Oxygen Species in Photocatalysis. *Chem Rev* 117, 11302-11336, doi:10.1021/acs.chemrev.7b00161 (2017).

In vitro measurements seem to show that the rods fully cover the neuron surface. In these conditions thermal effect might have a role to play, and it should be considered before being ruled out.

Response: We thank this reviewer for the constructive suggestion on thermal effect. As shown in the figure below, under 532 nm (23.5, 41.6 mW/cm²) and 670 nm (23.5, 41.6 mW/cm²) illumination, both the light source and ZST temperature changed less than 1 °C, which could not effectively stimulate neurons. Therefore, the interference of thermal effect on photocapacitive nerve stimulation of ZST can be excluded.

Extended Data Figure. S5 | The temperature changes over time for ZST under laser illumination at a wavelength of 532 nm with the power of (a) 23.5 and (b) 41.6 mW/cm², 670-nm with the power of (c) 23.5 and (d) 41.6 mW/cm².

1.42 “.higher stimulation voltage” is comparative with what?

Response: Thanks for the rigorous thinking by this reviewer. The “higher” comparison here refers to the glial scar caused by inflammation after long implantation of conventional electrodes. Due to the long-term implantation of electrodes leading to glial proliferation, a higher stimulus voltage is required to effectively induce neural activity compared to the initial implantation.

We have made revisions to clarify the content and eliminate the ambiguity that readers might have.

1.44 “reaction oxygen generation” is a mistyping

Response: We do apologize for the mistake. We used “reactive oxygen generation” to replace “reaction oxygen generation” in our revised manuscript.

1.52 the solution proposed here do need as well optical fibers to reach deep brain regions. This should be fully acknowledged and discussed.

Response: Thanks to this comment. By contrast to porphyrin monomer, the J-aggregated porphyrins exhibit a broad absorption band over visible light region, and an improved near-infrared two-photon characteristics. In our experiments, we succeeded in demonstrating that our nanoscale optoelectrodes in superficial cortex can be activated by extracranial 670 nm one-photon laser irradiation, without the need of optical fiber implantation (Fig. 4c-d).

In the behavior experiment, we observed the generation of hind limb movement in mice with the optoelectrodes pre-injected in the M2 region upon extracranial laser irradiation, either by one-photon process at 670 nm or by two-photon excitation at 850 nm (Fig. 4f-i). With this set of data, we believe, once the automation control of the two-photon laser beam can be realized to track the moving mice, our approach may allow high-spatial resolution modulation of neurons as well as fluorescence tracing *in vivo*, as the two-photon infrared light has deep tissue penetration due to the low linear absorption and scattering coefficient of biological specimen in the near-infrared range.

Since our optoelectrodes can activate the action potential of neurons upon ms pulsed light irradiation *in vitro*, we decided to test the effectiveness of our optoelectrodes *in vivo*. In clinical practice, high-frequency electrical stimulation has been found to be effective for Parkinson's disease (PD) treatment. So we constructed the PD model mice and applied the 130 Hz photoelectronic treatment for one week. We observed that this treatment can reduce the spontaneous firing activity of neurons in STN, thus preventing excessive activity and abnormal rhythmic burst firing. Furthermore, it has been observed this treatment increases the number of dopaminergic neurons' functional units, indicating that high-frequency stimulation might promote neurogenesis. All these results indirectly reflect the effectiveness of our optoelectrodes in neuromodulation *in vivo*, showing the potential to guide the exploration of the biological mechanisms in the treatment of neurological disorder.

Fig. 4 | *In vivo* neuromodulation by extracranial one-photon or two-photon laser stimulation. **a**, The 3D two-photon confocal fluorescence imaging of ZST (red fluorescence) and neurons (green fluorescence) in the mice genetically-encoded by GCaMP6 probes, with ZST pre-injected in M2. **b**, Bio-TEM image of ZST distribution in tissues 30 days later after ZST was injected into brain (the yellow arrows point to ZST). **c-d**, *In vivo* multichannel electrophysiology used to monitor the neuronal firing frequency of mice under different treatments. (-Laser: no laser; +Laser: laser irradiation; -ZST: no ZST; +ZST: ZST pre-injected in M2). The results are shown as mean±SD by two way ANOVA by Bonferroni's post hoc test, N=7 neurons from 3 mice in each group. **e**, The scheme of mice movement test. **f-g**, The movement at hind climb in mice with or without ZST pre-injected in M2, combined with or without extracranial 670 nm laser stimulation. The results are shown as mean±SD by two way ANOVA by Bonferroni's post hoc test, N=3 mice. **h-i**, The movement at hind climb in mice with or without ZST pre-injected in M2, combined with or without extracranial 850 nm fs laser stimulation. The results are shown as mean±SD by two way ANOVA analysis followed by Bonferroni's post hoc test, N=3 mice.

1.66 “surface, achieving direct charges injection in electrolyte via a capacitive process”. A capacitive process does not inject charge carriers, by definition.

Response: Thanks for the rigorous thinking by this reviewer. The “charge injection” we describe is intended to express “the ion current formed by the capacitive charges of ZST-electrolyte interface”.

To describe more accurately, in the revised draft, we have revised this sentence as follows: “surface, achieving ion current flow at ZST-electrolyte interface via a capacitive process”.

l.79 “single-photon” is misleading, it should be one-photon

Response: We do apologize for the mistake. We used “one-photon” to replace “single-photon” in our revised manuscript.

Minor comments:

1. L. 79 Non-Linear Optics is usually abbreviated as NLO, not “NOL”.

Response: We do apologize for the mistake. We used “NLO” to replace “NOL” in our revised manuscript.

2. L. 112 “across section” is cross-section.

Response: We do apologize for the mistake. We used “cross-section” to replace “across section” in our revised manuscript.

Viability: (i) could the authors explain the choice to use MTT assay for testing the viability of neurons instead of FDA/PI staining? (ii) histograms reported in the Extended Figure S5 present differences across the experimental conditions. Did the authors perform statistical analysis on those results? Are those differences statistically and/or biologically significant?

Response: We thank this reviewer for raising the question.(i) Here, we chose MTT assay for testing the viability of neurons basing on the following consideration. (1) MTT assay is primarily used to for cell proliferation and toxicity testing. (2) ZST can be excited by 488 nm laser, emitting red fluorescence. PI dye as a dead cells label, can also emit red fluorescence under the excited by 488 nm laser. This can make it difficult to distinguish between the PI and the ZST, leading to false positives. So we chose MTT instead of FDA/PI staining.

(ii) We performed the statistical analysis about the histograms reported in the Extended Figure S7. The two-way repeated measures analysis of variance (ANOVA) and Bonferroni post hoc analyses were used. Data are shown as individual values or expressed as the mean \pm SEM and significance levels are indicated as * $p < 0.05$, ** $p < 0.01$, *** $p < 0.001$, and not significant (n.s.). Statistically, there were no differences in cell viability under various illumination times and ZST concentrations, indicating the excellent biocompatibility of ZST under its working conditions.

Extended Data Figure. S7 | The viability of cells with varied ZST concentrations and 532 nm laser irradiation times by a typical MTT assay

Figure 2: Labels 1 and 2 are swapped between text and plot.

Response: We do apologize for the mistake. The labels 1 and 2 have been swapped between text and plot.

Fig. 2 | g, The reflected laser excitation beam (label 1) and the excitation on the local sample area (label 2). h, The common Gaussian distributions of spot 1 and 2 in (g).

Figure 3: (i) it would be clearer for the readers to include a graph of the percentage of failure (of AP) inside the main figure. (ii) please change the colors of the conditions 0.5 and 2.0 ms (too similar).

Response: (i) We thank this reviewer for the constructive suggestions. We have already inserted the graph depicting the failure rate of action potentials into the main figure (Fig 3j). At 130 Hz, neurons start to fail in generating one action potential (AP) for every pulse of light, with approximately $76.4\% \pm 2.7\%$ of the APs failing. This outcome can be attributed to the fact that about 80%-90% of the cortical neurons cultured here are excitatory glutamatergic neurons with low firing frequencies in response to depolarizing stimulation. These neurons have membrane properties

that make them less sensitive to high-frequency stimuli.

Fig. 3 | j, The percentage of failed APs at 130 Hz that the neuron begins to fail to generate 1 AP per pulse of light from 5 independent neurons.

(ii) In the revised manuscript, we have changed the colors of the conditions 0.5, 2.0 ms and 3.5, 23.5 mW/cm² for more clear reading (Fig 3e-f).

Fig 3 | e-f, The generation of AP on neurons pre-treated by ZST with different 532 nm laser power and pulse duration illumination.

Figure 4: (i) Explain the choice not to investigate microglia besides astrocytes.

(ii) authors should provide information of the morphology of the astrocytes (not only the number) as an additional indication of the absence of astrogliosis.

Response: We thank this reviewer for the constructive suggestions.

(i) Given that both astrocytes and microglia play crucial roles as cellular mediators in both acute and chronic neuroinflammatory responses, collectively constituting the central nervous system's innate immune system, a comprehensive evaluation is essential. To provide a more comprehensive assessment of the ZST's biocompatibility, we conducted additional experiments involving immunostaining of microglial. Our observations indicate that microglial did not exhibit any changes in morphology across different treatment groups, including cell body size, number of processes, and total process length (Fig. S9c-d).

(ii) Furthermore, in response to the reviewer's suggestion, we conducted a morphological analysis of astrocytes. We also observed that astrocytes did not undergo any changes in morphology across

different treatments, including surface area, number of processes, and total process length (Fig. S9 a-b). These results suggest the excellent biocompatibility and safety of ZST for neuromodulation.

Extended Data Figure. S9 | a, The immunostaining of GFAP expressed in mice M2 that was collected 30 days after different treatments. **b**, The corresponding statistical numbers of process, total process length, and surface area of astrocytes. **c**, The immunostaining of IBA1 expressed in mice M2 that was collected 30 days after different treatments. **d**, The corresponding statistical numbers of process, total process length, and cell body size of microglia. The results are shown as mean \pm sem by two-way ANOVA analysis followed by Bonferroni's post hoc test ($n = 14$ cells in astrocytes; $n = 12$ cells in microglia; $N = 3$ mice in each group ($*p < 0.05$, $**p < 0.01$, $***p < 0.001$)).

Figure 6: is the increase in the sIPSC frequency related to a modification of the number of synaptic boutons and/or Ca^{2+} dynamics? Please comment on this and/or provide new data.

Response: We thank the reviewer for raising the question about the sIPSC frequency. In our experiment, we observed the increase in the sIPSC frequency but not the amplitude may be related to an increased probability of Ca^{2+} influx in presynaptic inhibitory neurons after 130 Hz photoelectric stimulation. High-frequency electrical stimulation can excite the terminals of inhibitory synapses, resulting in an increased probability of calcium ion influx, which, in turn, leads to an increased release rate of GABA neurotransmitters, ultimately increasing the frequency of inhibitory synaptic currents in STN. It's worth noting that there is a substantial inhibitory input from the Globus Pallidus Externa (GPe) in STN, with GABAergic terminals on the somata constituting approximately 60% of the total boutons¹⁷. Early studies have also reported that STN-HFS reduces the activity of STN neurons. Mantovani et al., demonstrate that high frequency stimulation (HFS) significantly increased basal GABA outflow in the presence of submaximal concentrations of the voltage-gated sodium channel opener veratridine. This effect could be abolished by the GABA antagonists bicuculline or picrotoxin¹⁸. Moreover, it was recently demonstrated that HFS has specific effects on GABAergic terminals resulting in an enhancement of extracellular GABA through a GABA transporter-dependent mechanism in rats¹⁹.

References:

17. Milosevic, L. et al. Neuronal inhibition and synaptic plasticity of basal ganglia neurons in Parkinson's disease. *Brain* 141, 177-190, doi:10.1093/brain/awx296 (2018).

18. Mantovani, M., Van Velthoven, V., Fuellgraf, H., Feuerstein, T. J. & Moser, A. Neuronal electrical high-frequency stimulation enhances GABA outflow from human neocortical slices. *Neurochem Int* 49, 347-350, doi:10.1016/j.neuint.2006.02.008 (2006).

19. Li, T. L., Qadri, F. & Moser, A. Neuronal electrical high-frequency stimulation modulates presynaptic GABAergic physiology. *Neurosci Lett* 371, 117-121, doi:10.1016/j.neulet.2004.08.050 (2004).

Materials and methods section:

(i) The authors did not report the level of purity of the neuronal culture. Are astrocytes present in primary cultured neurons? any idea about the percentage?

Response: We thank the reviewer for raising the question about the level of purity of the neuronal culture. To address this, we further conducted immunofluorescence experiments using specific protein markers on rat cortical cells cultured for 15 days in vitro. These results indicate that the neuronal purity is up to $75.8\% \pm 2.4\%$, the astrocyte purity is approximately $5.7\% \pm 0.3\%$, and the remaining $18.5\% \pm 2.3\%$ may be comprised of other types of glial cells, such as microglia (Figure S6).

Extended Data Figure. S6 | a, Purity of primary neuron cultures as evidenced by immunolabeling with NeuN (neuronal marker), GFAP (astrocyte marker), and DAPI (cellular nuclear marker). **b,** The average percentage of various cell types in primary cell cultures varies (neurons, $75.8\% \pm 2.4\%$; astrocyte, $5.7\% \pm 0.3\%$; others, $18.5\% \pm 2.3\%$; $n = 8$ plates of cells).

(ii) why the authors chose to use a pipette solution with high concentration of Cl^- ?

Response: We apologize for the mistake. We used a pipette solution with a low concentration of Cl^- in our experiments, and we have corrected this mistake in the revised manuscript.

(iii) why the authors avoided to addition of CGP (or other specific blocker of GABAB receptor) for the sPSC recordings? Please comment on this.

Response: Thanks for this reviewer's comments. Because the recorded inhibitory postsynaptic currents are predominantly mediated by GABAA receptors, which are responsible for fast inhibitory synaptic transmission. GABAA receptors belong to the ionotropic receptor class, signifying that they function as ligand-gated ion channels. Activation of GABAA receptors elicits a rapid and direct inhibitory response in the neuron. While GABAB receptors are metabotropic receptors, mediate slower, more prolonged inhibition through intracellular signaling pathways²⁰. The distinct dynamics of currents mediated by GABAA and GABAB receptors make them easily distinguishable without the need for GABAB receptor blockers.

References:

20. Shaye, H., Stauch, B., Gati, C. & Cherezov, V. Molecular mechanisms of metabotropic GABA receptor function. *Sci Adv* 7, doi:ARTN eabg336210.1126/sciadv.abg3362 (2021).

(iv) it lacks a paragraph explaining the statical analysis.

Response: Thanks for this reviewer's comments. In this revised manuscript, we add descriptions of the statistical analyses.

Reviewer #2 (Remarks to the Author):

The manuscript describes the design and validation of nanoparticles for neuromodulation using photoelectric materials. It details the synthesis of nanoparticles of ZS which are then coated with TiO₂ to produce ZST in the 100s of nm size range. Characterization of the nanoparticles in terms of optical properties validates its sensitivity to the given wavelengths and their efficiency. ZST were then placed with cultured cortical neurons and demonstrated their ability to elicit action potentials from light, a fast effect with low energy density that was dependant on the nanorods. The particles were then injected into mice brains and activated with light to stimulate hindlimb activity without a distinct immune response. This was elicited with both single photon at 670 nm or two-photon at 850 nm. They were then used as a possible treatment for PD in mice. Overall, the manuscript is convincing for ZST as an effective acute light-based wireless neuromodulation device.

Major comments:

Localization of the ZST was validated after 30 days however their function was not addressed during this month. It begs the question of whether the nanorods were functional, and to what degree, during the 1-month chronic injection used to produce this histology. An assessment of at least short-term ageing in *in vitro* or *in vivo* conditions would address this concern.

Response: We express our gratitude to this reviewer for recognizing the key innovations of our work. To gather further compelling evidence supporting the clinical applicability of our optoelectrode, we conducted additional experiments to confirm the long-term effectiveness of ZST functionality. After injecting ZST into the motor cortex, we utilized *in vivo* multi-channel techniques to record neuronal activities on the 7th, 15th, and 30th days, respectively. The results demonstrated that under 670 nm laser irradiation, ZST could activate neuronal firing even up to 30 days, with no significant decrease in the neuronal firing rates, indicating the sustained functionality of ZST without deterioration.

Extended Data Figure. S10 | a, *In vivo* multichannel electrophysiology used to monitor the neuronal firing of mice under different treatments on the 7th, 15th, and 30th days after ZST injected in M2. (-Laser: no laser; +Laser: laser irradiation; -ZST: no ZST; +ZST). b, The results are shown

as mean \pm SD by the two-way repeated measures analysis of variance (ANOVA) by Bonferroni's post hoc test, N=6 neurons from 3 mice in each group.

The inclusion of the Parkinson's disease section is interesting however feels out of place given the main goal of the article is to present a novel technology for neuromodulation. Firstly, it requires the implantation of an optical fibre for use which negates the main benefit of ZST as untethered. Critically, the protocol presented does not seem to have been used in previous literature as a treatment so an equivalent electrical device may explain why this stimulation protocol (10 days of 3 minutes of pulse trains prior to behavioural tests) was chosen and how it was known that it would improve PD symptoms. In the given reference, stimulation is provided during the behavioural tests and not as a longer-term treatment. In this manuscript, it seems like the stimulation regrows neurons so that function is regained and is present even when the light stimulation is not continued. This is a very interesting result but needs more exploration. This section needs to be expanded to be convincing or should be incorporated in a separate article.

Response: We thank the reviewer for raising the questions. Since our optoelectrodes can activate the action potential of neurons upon ms pulsed light irradiation *in vitro*, we decided to test the effectiveness of our optoelectrodes *in vivo*. In our experiments, we succeeded in demonstrating that our nanoscale optoelectrodes in superficial cortex can be activated by extracranial 670 nm one-photon laser irradiation, without the need of optical fiber implantation (Fig. 4c-d).

In the behavior experiment, we observed the generation of hind limb movement in mice with the optoelectrodes pre-injected in the M2 region upon extracranial laser irradiation, either by one-photon process at 670 nm or by two-photon excitation at 850 nm (Fig. 4f-i). With this set of data, we believe, once the automation control of the two-photon laser beam can be realized to track the moving mice, our approach may allow high-spatial resolution modulation of neurons as well as fluorescence tracing *in vivo*, as the two-photon infrared light has deep tissue penetration due to the low linear absorption and scattering coefficient of biological specimen in the near-infrared range.

In clinical practice, high-frequency electrical stimulation has been found to be effective for PD treatment. We chose a 10-day treatment duration for long-term Parkinson's disease therapy because effectively managing neurodegenerative diseases requires continuous modulation of neuronal function, which necessitates extending the stimulation period. This approach aligns with findings by Welter, M. L. et al., who demonstrated that a single 130Hz stimulation promptly enhances motor function in mice with PD, indicating an immediate effect from singular stimulation. However, for achieving persistent and reliable treatment for PD, the cumulative impact of repeated stimulations is crucial, leading to enduring and stable plastic changes. Wang, X.Y., et al., using optogenetics, discovered that a single stimulation can activate the DRN→VTA neural circuit and modulate the behavior of mice. However, their observations show that continuous activation of the DRN→VTA

neural circuit over 14 days effectively and consistently alleviated pain and depressive behavior in mice²¹.

So we have opted for a 10-day regimen of long-term stimulation for the treatment of PD in mice. We observed that this treatment can reduce the spontaneous firing activity of neurons in STN, thus preventing excessive activity and abnormal rhythmic burst firing. Furthermore, it has been observed this treatment increases the number of dopaminergic neurons' functional units, indicating that high-frequency stimulation might promote neurogenesis. All these results indirectly reflect the effectiveness of our optoelectrodes in neuromodulation *in vivo*, showing the potential to guide the exploration of the biological mechanisms in the treatment of neurological disorders.

Indeed, the increase in dopamine neurons in the substantia nigra of PD mice in our experiment is a very promising result. It could be due to the regenerative effects of high-frequency electrical stimulation, or on the other hand, it might be the deep brain stimulation promoting the release of BDNF to protect the dopamine neurons in PD mice, preventing their continued loss. This is a fascinating phenomenon that warrants further exploration, and we will continue to investigate the underlying biological mechanisms.

Fig. 4 | *In vivo* neuromodulation by extracranial one-photon or two-photon laser stimulation. a, The 3D two-photon confocal fluorescence imaging of ZST (red fluorescence) and neurons (green fluorescence) in the mice genetically-encoded by GCaMP6 probes, with ZST pre-injected in M2. **b,** Bio-TEM image of ZST distribution in tissues 30 days later after ZST was injected into brain (the yellow arrows point to ZST). **c-d,** *In vivo* multichannel electrophysiology used to monitor the neuronal firing

frequency of mice under different treatments. (-Laser: no laser; +Laser: laser irradiation; -ZST: no ZST; +ZST: ZST pre-injected in M2). The results are shown as mean±SD by two way ANOVA by Bonferroni's post hoc test, N=7 neurons from 3 mice in each group. **e**, The scheme of mice movement test. **f-g**, The movement at hind limb in mice with or without ZST pre-injected in M2, combined with or without extracranial 670 nm laser stimulation. The results are shown as mean±SD by two way ANOVA by Bonferroni's post hoc test, N=3 mice. **h-i**, The movement at hind limb in mice with or without ZST pre-injected in M2, combined with or without extracranial 850 nm fs laser stimulation. The results are shown as mean±SD by two way ANOVA analysis followed by Bonferroni's post hoc test, N=3 mice.

References:

21. Wang, X. Y. et al. A glutamatergic DRN-VTA pathway modulates neuropathic pain and comorbid anhedonia-like behavior in mice. *Nat Commun* 14, 5124, doi:10.1038/s41467-023-40860-3 (2023).

Minor comments:

This is a photocapacitive device, not photovoltaic.

Response: We do apologize for the mistake. We used “photocapacitive” to replace “photovoltaic” in our revised manuscript.

Not all faradaic processes are irreversible. Reversible redox reactions are common in current neural devices.

Response: We thank this reviewer for the constructive suggestion. There is indeed a reversible Faraday process in electrical nerve stimulation by switching the positive and negative current directions. For more accurate description, we have corrected this relevant expression in our revised manuscript.

Figure 1c is not clear in describing the working principle of the device as it appears charges enter the tissue, as opposed to charging a capacitor.

Response: We thank this reviewer for raising the question. There may be some misunderstanding about the working principle of the device. The red areas in Figure 1b and Figure 1c refer to the TiO₂, but not the tissue. For a clear show, we added a text description in our revised Figure 1.

Fig. 1 | The non-Faradaic capacitive photovoltaic mechanism of ZST for optoelectrical modulation of neurons. **a**, Random aggregation of ZnTPyP results in serious intramolecular exciton-exciton annihilation during exciton migration. **b-c**, J-aggregated ZnTPyP array enables the long-range exciton diffusion for ultrafast electron transfer into TiO₂, forming electron-rich TiO₂(e⁻). Meanwhile, the co-generated holes are eliminated by the reductive scavengers in a physiological environment like GSH for dye regeneration. i, exciton migration; ii, hole-electron pair separation; iii, electron transfer.

In Figure 3b, it is unclear on the location of the ZST. An outline of the nanorods would help.

Response: We thank this reviewer for raising the question about the location of the ZST. As shown from the SEM images, ZST was loaded outside the membrane in the region outlined by the red dashed line

Fig. 3b | SEM images of ZST within the region outlined by the red dashed line on the outer membrane of neurons.

For the acute hindlimb experiments, was the skin closed?

Response: We thank this reviewer for raising the question. The scalp of the mouse was not sutured in this acute hindlimb experiments.

In the videos, it appears as though there is some residual movement when the light is off. With the 850 nm video, the red dot timing is sometimes off as the muscle twitches first.

Response: We thank this reviewer for raising the question. The residual movement in the hind limbs may be due to the sustained stimulation, which makes the neurons hard for total recovery immediately. So we can see the minor hind limb movements before subsequent 850 nm laser irradiation.

REVIEWERS' COMMENTS

Reviewer #1 (Remarks to the Author):

I am satisfied with the changes, and I believe the paper is now suitable for publication.

Reviewer #2 (Remarks to the Author):

For the chronic stability, thank you for addressing my concern.

For the PD section, the authors rely on novel innovation in the space of two photon automation for tracking. In this case, there does not seem any added benefit compared to optogenetic techniques which can provide similar if not better spatial control as, like the ZSTs, they are injected.

The justification for the treatment protocol still seems lacking as with the references provided, the stimulation was provided during, not before, assessment of the PD symptoms in mice. Why was a test with single high frequency stimulation during the open field test not conducted? For Welter M L et al (2004), this was done in humans so this sentence is wrong: "This approach aligns with findings by Welter, M. L. et al., who demonstrated that a single 130Hz stimulation promptly enhances motor function in mice with PD"

The Wang X Y et al paper also does not show "that continuous activation of the DRN→VTA neural circuit over 14 days effectively and consistently alleviated pain and depressive behavior in mice" like was stated in the rebuttal and does not relate to improvement of PD symptoms without stimulation.

Nonetheless, the 10 day regimen does show evidence of improvement of symptoms and regeneration of dopaminergic neurons which is very interesting. More experiments would be needed to further validate this and make the neural circuitry claim of Fig 6 about what the ZST is stimulating and its effect.

You state "All these results indirectly reflect the effectiveness of our optoelectrodes in neuromodulation in vivo" which was already true from the motor cortex stimulation but the claims into neural circuitry are not fully proven to the point of convincing inclusion in this article.

Minor comments were satisfactorily addressed.

Another comment: In the methods for gait analysis, it says "rats that were visually observed..." but the experiment used mice.

REVIEWERS' COMMENTS

Reviewer #2 (Remarks to the Author):

For the PD section, the authors rely on novel innovation in the space of two photon automation for tracking. In this case, there does not seem any added benefit compared to optogenetic techniques which can provide similar if not better spatial control as, like the ZSTs, they are injected.

Response: Thanks for the rigorous thinking by this reviewer. To address the reviewer's concerns regarding the distinctive advantages of our method, we have conducted a comprehensive review and analysis of the advancements provided by our technique when compared to conventional optical fiber-based optogenetics. Optogenetics has advanced the field by spatiotemporal precision, however, it requires the use of genetically encoded and optically active proteins to control neuronal activity, which limits its translational potentials. Usually, the genetic modification process needs 2-4 weeks to express the target protein in neurons, thus increasing the cost of time and the probability of failure. Moreover, the photosensitive proteins are activated by a certain visible light at a power density of ~300-700 mW/cm², which may damage the nerve due to the thermal effect of the excitation light. In contrast, our technique eliminates the need for gene transfection, providing adjustable nano-sized optoelectrodes suitable for a wide range of *in vitro* and *in vivo* applications, with improved technical convenience. The broadened excitation light wavelength from visible to NIR enhances versatility for neuromodulation, with a low work power density ensuring high biosafety. To enhance clarity and reference, we have summarized these essential details in a tabular format.

Table 1. The comparison of nanoscale optoelectrodes with optogenetics.

Differences \ Types	Optogenetics	Our optoelectrodes
Pre-treatment on neurons	Gene transfection for 2-4 weeks	In vitro co-culturing for 3-6 hours; in vivo microinjection for 3 days
Laser power density	300-700 mW/cm ²	3-25 mW/cm ²
Laser wavelength	Visible light (400-700 nm)	One-photon excitation (400-700 nm) Two-photon excitation (850 nm fs NIR laser)

Stimulation safety	Thermal effect causing phototoxicity	No photo-electrochemical damage to neurons and no thermal effect
--------------------------------------	--

The justification for the treatment protocol still seems lacking as with the references provided, the stimulation was provided during, not before, assessment of the PD symptoms in mice. Why was a test with single high-frequency stimulation during the open field test not conducted? For Welter M L et al (2004), this was done in humans so this sentence is wrong: "This approach aligns with findings by Welter, M. L. et al., who demonstrated that a single 130Hz stimulation promptly enhances motor function in mice with PD" The Wang X Y et al paper also does not show "that continuous activation of the DRN→VTA neural circuit over 14 days effectively and consistently alleviated pain and depressive behavior in mice" like was stated in the rebuttal and does not relate to the improvement of PD symptoms without stimulation.

Response: We thank this reviewer for raising the question around the treatment protocol for PD. To address reviewer's concern regarding the rationale of the treatment protocol, we have summarized some literature on the long-term deep brain stimulation (DBS) for the treatment of PD (Table 2). The choice of 10 days treatment duration in DBS for PD is grounded in the understanding of PD as a progressive neurological disorder characterized by its chronic and evolving nature, necessitating ongoing symptom management. Continuous stimulation becomes crucial for sustained relief and improved brain function with PD. Caryl E. Sortwell et al. discovered that STN-DBS, administered at a frequency of 130 Hz, pulse width of 60 μ s, and intensity ranging from 30 to 50 μ A for 2 weeks, was effective in treating the motor symptoms of PD and preventing the loss of dopaminergic neurons^{1,2}. Won Jong Kim et al. also observed a gradual improvement in motor functions through ultrasound-mediated piezoelectric DBS applied to the STN, employing daily sequential stimulation for 10 days³. Their findings suggest that piezoelectric stimulation potentially alleviates the symptoms of PD by inhibiting the degeneration of dopaminergic neurons and inducing the release of dopamine. Furthermore, Lei Sun et al. discovered that sonogenetic stimulation of the STN in PD model mice for five consecutive days improved their motor coordination and mobility⁴. These findings indicate that long-term DBS is more effective in consistently improving the symptoms of PD.

Table 2. The treatment protocol for Parkinson’s disease

	Stimulation parameters	Stimulation duration	Reference
Metal electrode	130 Hz, 60 μ s pulse width, 30–50 μ A	2 weeks	Neurobiol Dis 39, 105-115 (2010)
Metal electrode	130 Hz, 60 μ s pulse width, 30–50 μ A	1-2 weeks	J Neurosci 37, 6786-6796 (2017)
Metal electrode	130 Hz, 60 μ s, pulse width	21 days	Ann Neurol 81, 825-836 (2017)
Piezoelectric nanoparticle	1.5 MHz, 462.4 W cm ⁻² , 10% duty cycle, 10 Hz pulse repetition frequency, 60 s per day	10 days	Nat Biomed Eng 7, 14 (2023).
Mechanosensitive ion channel (MscL-G22S)	0.5 MHz or 0.9 MHz central frequency, 1 kHz PRF, 40% or 50% duty cycle, under burst mode at 300 ms duration, 30 min/day	5 days	PNAS , 120 (2023).
Electromagnetic - Nanoparticle	0.64 W cm ⁻² , 1 MHz, 3 min/day	7 days	Adv Mater 32 (2020)
Our optoelectrodes	130 Hz, trains of 6 trials (each trial was composed of 10 s on and 20 s off), 3 min/day	10 days	Our work

References:

1. Spieles-Engemann, A. L. et al. Stimulation of the rat subthalamic nucleus is neuroprotective following significant nigral dopamine neuron loss. *Neurobiol Dis* 39, 105-115 (2010).

2. Fischer, D. L. et al. Subthalamic Nucleus Deep Brain Stimulation Employs trkB Signaling for Neuroprotection and Functional Restoration. *J Neurosci* 37, 6786-6796 (2017).
3. Kim, T. et al. Deep brain stimulation by blood-brain-barrier-crossing piezoelectric nanoparticles generating current and nitric oxide under focused ultrasound. *Nat Biomed Eng* 7, 14 (2023).
4. Xian, Q. X. et al. Modulation of deep neural circuits with sonogenetics. *P Natl Acad Sci USA* 120 (2023).

Nonetheless, the 10-day regimen does show evidence of improvement of symptoms and regeneration of dopaminergic neurons which is very interesting. More experiments would be needed to further validate this and make the neural circuitry claim of Fig 6 about what the ZST is stimulating and its effect. You state "All these results indirectly reflect the effectiveness of our optoelectrodes in neuromodulation in vivo" which was already true from the motor cortex stimulation but the claims into neural circuitry are not fully proven to the point of convincing inclusion in this article.

Response: We thank this reviewer for raising the question regarding the neural circuit mechanisms of PD treatment. Currently, DBS is the gold-standard neurosurgical therapy for PD with the STN as the most commonly targeted and studied site. Excitingly, in our experiments, we also observed an improvement in the motor function of PD mice by stimulating the STN using our optoelectrodes. Furthermore, we explored that the improvement in Parkinson's symptoms may be attributed to the increase in dopaminergic neurons and the reduction in the excitability of STN, consistent with findings from previous research^{1,2,3}. Specifically, the observed increase in dopaminergic neurons could be attributed to the protective effect of optoelectrical stimulation of the STN against the ongoing loss of dopaminergic neurons. Early studies have already indicated that STN-DBS protects dopaminergic neurons by attenuating neuronal cell death to limit synaptic dysfunction in neurodegenerative disorders, thereby slowing the disease progression including deficits of motor function, cognitive and fear conditioning⁴. They found that STN-DBS can promote neuronal survival by stimulating the release of BDNF^{1,2,5}. Following the release, BDNF binds the transmembrane receptor tropomyosin-related kinase type B (trkB) resulting in the activation of three intracellular signaling cascades: (1) mitogen-activated protein kinase/extracellular signal related-kinase which promotes protein synthesis; (2) phosphatidylinositol 3-kinase/protein kinase B which regulates protein translation/trafficking and inhibits apoptosis; (3) phospholipase C/protein kinase C which is involved in the regulation of synaptic plasticity⁶. Taken together, these findings identify

BDNF/trkB signaling as a possible mechanism for STN-DBS mediated neuroprotection.

The reduction in excitability of the STN may be another reason for the improvement of PD symptoms. The increased activity in the STN is considered to cause symptoms of PD, including akinesia and rigidity⁷. Some studies have reported that lesions in or pharmacological blockade of the STN were found to significantly improve motor symptoms in PD^{8,9}. Furthermore, HFS of the STN was later reported to be highly effective in reversing PD symptoms, producing an effect similar to inducing lesions or pharmacologically blocking the STN¹⁰. This may be because high-frequency electrical stimulation can excite the terminals of inhibitory synapses, resulting in an increased release rate of GABA neurotransmitters, in turn, decreasing the activity in the STN. It was recently demonstrated that HFS has specific effects on GABAergic terminals resulting in an enhancement of extracellular GABA through a GABA transporter-dependent mechanism¹¹. Mantovani et al. also demonstrated that HFS significantly increased basal GABA outflow in the presence of submaximal concentrations of the voltage-gated sodium channel opener veratridine¹². This effect could be abolished by the GABA antagonists bicuculline or picrotoxin¹². In our work, we also found an increase in the frequency of IPSCs leads to the reduced excitability of the STN. It may be because our optoelectrical stimulation can increase the release rate of presynaptic GABA neurotransmitters.

In summary, the positive effect of our optoelectrodes in improving PD symptoms may be attributed, on one hand, to the activation of the BDNF/trkB signaling pathway, leading to an increase in dopaminergic neurons, and on the other hand, to the increase in the frequency of IPSC that reduces the excitability of the STN. There may be other mechanisms at play, and we will further explore them.

References:

1. Spieles-Engemann, A. L. et al. Stimulation of the rat subthalamic nucleus is neuroprotective following significant nigral dopamine neuron loss. *Neurobiol Dis* 39, 105-115 (2010).
2. Fischer, D. L. et al. Subthalamic Nucleus Deep Brain Stimulation Employs trkB Signaling for Neuroprotection and Functional Restoration. *J Neurosci* 37, 6786-6796 (2017).
3. Welter, M. L. et al. Effects of high-frequency stimulation on subthalamic neuronal activity in parkinsonian patients. *Arch Neurol-Chicago* 61, 89-96, (2004).
4. McKinnon, C. et al. Deep brain stimulation: potential for neuroprotection. *Ann Clin Transl Neurol* 6, 174-185 (2019)
5. Spieles-Engemann, A. L. et al. Subthalamic Nucleus Stimulation Increases Brain Derived

Neurotrophic Factor in the Nigrostriatal System and Primary Motor Cortex. *J Parkinson Dis* 1, 123-136 (2011).

6. Yoshii, A. & Constantine-Paton, M. Postsynaptic BDNF-TrkB Signaling in Synapse Maturation, Plasticity, and Disease. *Dev Neurobiol* 70, 304-322 (2010).
7. Bergman, H., Wichmann, T., Karmon, B. & DeLong, M. R. The primate subthalamic nucleus. II. Neuronal activity in the MPTP model of parkinsonism. *J Neurophysiol* 72, 507-520 (1994).
8. Bergman, H., Wichmann, T. & DeLong, M. R. Reversal of experimental parkinsonism by lesions of the subthalamic nucleus. *Science* 249, 1436-1438 (1990)
9. Wichmann, T., Bergman, H. & DeLong, M. R. The primate subthalamic nucleus. III. Changes in motor behavior and neuronal activity in the internal pallidum induced by subthalamic inactivation in the MPTP model of parkinsonism. *J Neurophysiol* 72, 521-530 (1994).
10. Benazzouz, A., Gross, C., Feger, J., Boraud, T. & Bioulac, B. Reversal of rigidity and improvement in motor performance by subthalamic high-frequency stimulation in MPTP-treated monkeys. *Eur J Neurosci* 5, 382-389 (1993).
11. Li, T. L., Qadri, F. & Moser, A. Neuronal electrical high-frequency stimulation modulates presynaptic GABAergic physiology. *Neurosci Lett* 371, 117-121.
12. Mantovani, M., Van Velthoven, V., Fuellgraf, H., Feuerstein, T. J. & Moser, A. Neuronal electrical high-frequency stimulation enhances GABA outflow from human neocortical slices. *Neurochem Int* 49, 347-355.

In the methods for gait analysis, it says "rats that were visually observed..." but the experiment used mice

Response: We do apologize for the mistake. We used "mice" to replace "rats" in our revised manuscript.